# *"It was very nice to be in a room where everyone had ADD—that's kind of VIP"*: Exploring clients' perceptions of group CBT for ADHD inattentive presentation

Elinor Eskilsson Strålin[1]*, Rikard Sunnhed[1], Lisa B. Thorell[2], Tobias Lundgren[1], Sven Bölte[3,4,5], Benjamin Bohman[1]

1 Department of Clinical Neuroscience, Karolinska Institutet, & Stockholm Health Care Services, Centre for Psychiatry Research, Region Stockholm, Stockholm, Sweden, 2 Department of Clinical Neuroscience, Karolinska Institutet, Stockholm, Sweden, 3 Department of Women's and Children's Health, Karolinska Institutet, Region Stockholm, Center of Neurodevelopmental Disorders (KIND), Centre for Psychiatry Research, Stockholm Health Care Services, Stockholm, Sweden, 4 Child and Adolescent Psychiatry, Stockholm Health Care Services, Region Stockholm, Stockholm, Sweden, 5 Curtin Autism Research Group, Curtin School of Allied Health, Curtin University, Perth, Australia

* elinor.eskilsson.stralin@ki.se

**Data Availability Statement:** Data cannot be shared publicly because of Swedish legal and ethical restrictions related to sensitive patient

## Abstract

### Objectives

This qualitative study explored the perceptions of adult clients participating in a new psychological treatment for attention-deficit/hyperactivity disorder inattentive presentation (ADHD-I, also called "attention deficit disorder", ADD). The study aimed to explore (i) what aspects of treatment the participants found to be helpful, and (ii) if there were areas that ought to be developed to make the protocol more useful to clients with ADHD-I.

### Methods

Participants were recruited from treatment groups following the protocol of cognitive-behavioral therapy for ADHD-I (CADDI), at three psychiatric outpatient units in Stockholm, Sweden. Individual semi-structured interviews, lasting on average 44,6 minutes, were conducted with 14 adults after the completion of CADDI. Interviews explored participants' perceptions of CADDI, its usefulness, and asked for suggestions for improvement. Interviews were conducted by independent interviewers and were transcribed verbatim. The text was analyzed using reflexive thematic analysis.

### Results

The analysis generated three themes: "Factors of importance for change", with the subthemes; the group, therapeutic components, structure of treatment, and motivation,"Gains in treatment", with the subthemes; insight and understanding, increased attention, and planning and acting, and "Challenges with ADHD-I and remaining needs", with the subthemes; ADHD as a lifelong condition, maintaining gains in treatment, and wish for further support.

information. Specifically, the participants did not consent to public data sharing. Data are available from Region Stockholm (contact via data protection officer Camilla Heise Löwgren, camilla. heiselowgren@regionstockholm.se) for researchers who meet the criteria for access to confidential data.

**Funding:** The study was funded by the Mental Health Foundation and Professor Bror Gadelius Foundation. Open access funding was provided by Karolinska Institutet, Stockholm, Sweden. EES received a grant from Mental Health Foundation https://www.fondenpsykiskhalsa.se/?gclid=

## Conclusions

Participants emphasized the importance of the group setting as a facilitator of therapeutic effects of increased understanding and self-acceptance. Further, they considered the practice of mindfulness to enhance attention and awareness of thoughts, feelings, and activities and considered the structure of treatment as supporting the work in therapy. These findings support the value of the group setting and confirm the usefulness of CADDI. However, participants were concerned about how to maintain gains of treatment after its termination and suggested follow-up sessions to improve the CADDI protocol.

## Trial registration

Preregistered at Clinical Trials: NCT05037903.

## Introduction

Attention-deficit/hyperactivity disorder (ADHD) is a persistent and heterogeneous neurodevelopmental condition operationalized by three phenotypic presentations in DSM-5 and ICD-11: predominantly inattentive (ADHD-I, also called "attention deficit disorder", ADD), predominantly hyperactive/impulsive, and combined presentation [1,2]. ADHD is common, affecting about 2–4% of adults worldwide [3]. ADHD is associated with negative consequences for health, social life, and productivity; therefore it is paramount to find ways to decrease the burden of living with the condition [4]. Both somatic and psychiatric conditions are commonly co-occurring in ADHD, stressing extended health care needs [4,5]. Adult ADHD is also associated with poor academic, occupational, and economic outcomes [6–8].

Multimodal treatments including psychoeducation, pharmacotherapy, and cognitive-behavioral therapy (CBT) are recommended in international guidelines for ADHD in adulthood [4]. CBT is administered using structured treatment protocols, focusing on the acquisition of compensatory skills to cope with common difficulties of ADHD. Meta-analyses of CBT trials for ADHD in adults indicate amelioration of ADHD symptoms and provide modest support for the use of CBT in ADHD [9–11]. The most studied and used CBT protocols for adults have been developed by Safren and colleagues [12,13] and Hesslinger and colleagues [14]. The Safren protocol focuses on strategies for organizing activities, to cope with distractibility and cognitive restructuring. The protocol has proved beneficial in reducing ADHD symptoms [12,13,15]. The Hesslinger protocol is an adaptation of dialectical behavior therapy to suit ADHD and uses mindfulness in conjunction with behavioral analysis to address symptoms and problem behaviors of ADHD. It has proven feasible, reduces symptoms of ADHD and is considered effective by participants [16–19]. However, the Hesslinger protocol focuses less on organizational skills and therefore lacks important components to treat difficulties associated with inattention [18].

Symptoms of inattention and hyperactivity/impulsivity are, in different ways, associated with impairment in daily life. Inattention in ADHD involves difficulties in organizing, starting and finishing activities, time management, distractibility and forgetfulness, among others [4]. Inattention is highly correlated with academic and occupational underachievement and is a robust predictor of long-term impairment [7,20]. Studies have found high levels of general stress, emotional and relational difficulties to be more closely associated with inattention than hyperactivity/impulsivity in adults [7,21–23]. Therefore, inattention features of ADHD deserve

specific focus in treatment strategies, such as CBT [20,24]. Moreover, previous research has shown that treatment of inattention should preferably be comprehensive and multifaceted, including several components and strategies [20,21]. So far, no CBT protocol has been developed to address inattention and associated organizational difficulties using a broad range of interventions. To address the impairing consequences of inattention using CBT, it seems necessary to include components of both organizational skills, mindfulness and behavioral activation to deal with procrastination and passivity in clients with ADHD-I [25,26]. Based on these observations, the cognitive-behavioral therapy for ADHD-I (CADDI) protocol was developed. Further, clinical and previous observations gave reason to believe that adults with ADHD-I would benefit from practicing and rehearsing organizational skills over time in treatment and that individual support in addition to group treatment might increase the benefits of the intervention [18,27]. The feasibility of CADDI has been evaluated in outpatient psychiatric care, including groups at four locations. The protocol demonstrated good feasibility, acceptability, and preliminary effects regarding inattentive and cooccurring depressive symptoms [27]. The objective of the current study was to explore participants' perceptions of CADDI to better understand how participants experience the intervention and their opinions on how it might be further improved. Participants were asked to reflect on their perceptions of the treatment after completion. The exploration was guided by the following research questions; i) What aspects of treatment are helpful according to participants? and ii) Are there areas that ought to be developed to make the protocol more useful to clients with ADHD-I?"

## Materials and methods

### Study design

We used qualitative method to inquire into our research questions and collected data through semi-structured interviews with individuals who had received treatment according to the CADDI protocol. Participants were recruited from an ongoing multicenter randomized controlled trial (RCT) of CADDI, conducted in psychiatric services in Stockholm, Sweden. The trial and the current study were approved by the Swedish Ethical Review Authority (2019–02444 and 2021–01663, respectively) and preregistered at Clinical Trials (NCT04090983 and NCT05037903, respectively).

### Participants

Recruitment started in June, 2021 and went on until June, 2022. Eligible participants in the RCT were adults with ADHD-I aged 18 years or older. Inclusion criteria were: (i) No change in medication the last two months and (ii) completion of a psychoeducational course on ADHD regarding symptoms, self-care and treatments. Exclusion criteria were: (i) severe mental illness (e.g., severe depression), (ii) substance abuse, or (iii) intellectual disability. Additional inclusion criteria were: (i) randomized to CADDI and (ii) completion (i.e. not dropping out) of the CADDI intervention. After completion of the CADDI protocol, all participants in three groups at separate locations, were informed about the study and invited to the interviews. All 15 accepted to participate and gave their informed consent in writing. Fourteen of them managed to schedule an interview for the study. In the participant group, nine were females (64%), and mean age was 32.6 years (SD = 8.2, range = 22–50) and average age when first diagnosed with ADHD-I was 29.9 years (SD = 9.1, range = 18–48). Nine (64%) participants were treated with stimulants with or without other pharmacotherapy, three (21%) were medication free and two (14%) were treated with non-stimulant medication. Regarding level of education, nine (64%) participants had completed high school, four (29%) had a bachelor's or master's degree and one (7%) participant had not completed high school. In the RCT there was 53

completers of CADDI, of them 34 (64%) were females, they had a mean age of 35.3 (SD = 8.4, range 21–53) years, and mean age when first diagnosed was 31.6 years (SD = 9.5 range 12–52); thus, the sample of the current study was similar to all completers of the CADDI protocol in the RCT.

## The CADDI protocol

The protocol was inspired by the work on psychological treatment of ADHD by Hesslinger and colleagues [28] and Safren and colleagues [12,13], as well as by the work on behavioral activation by Addis and Martell [29]. The CADDI protocol includes components regarding organization of activity, daily routines, coping with procrastination and negative affect, and uses behavior analysis to identify reinforcement contingencies. Mindfulness meditation is practiced during the sessions and as home assignments, based on audio-recordings of meditation exercises. The aim of mindfulness meditation in treatment of ADHD is to enhance attention and awareness of thoughts, feelings, and impulses to facilitate behavior change [30,31]. The content of the CADDI protocol is summarized in Table 1.

The CADDI protocol was delivered in a group format consisting of 14 weekly 2-hour sessions including a break and was designed for groups of 6 to 10 participants. The CADDI protocol was structured to provide support to overcome difficulties with inattention that might interfere with adherence to treatment. Therefore, home assignments are followed-up in weekly telephone calls by a group leader offering support and monitoring treatment progress. Participants are encouraged to share the content of each session with a significant other to enhance learning and involve close ones in the treatment. All components in the protocol are rehearsed over two sessions and followed-up continuously in the group. Some sessions are dedicated solely to repetition to enhance acquisition of new habits and routines. After each session, there is time for an optional short individual consultation regarding treatment progress.

**Table 1. Content of the cognitive-behavioral therapy for ADHD inattentive presentation (CADDI) protocol.**

| Session | Content |
|---|---|
| 1. Introduction | Introduction to treatment and basic organizational tools |
| 2. Prioritizing and Mindfulness | How to prioritize, introduction to mindfulness |
| 3. Routines in daily living 1 | Using a week-planner, making routines for recurring activities, mindfulness. |
| 4. Routines in daily living 2 | Weekly planning and how to divide big tasks into smaller pieces, mindfulness. |
| 5. Repetition and troubleshooting | Repetition of previous sessions. Setting goals and defining action in accordance with goals. Mindfulness. |
| 6. Getting started 1 | Skills for activity initiation, introduction to behavior analysis and weekly report chart to support behavior change. |
| 7. Getting started 2 | Follow-up on report chart for activity initiation, behavior analysis of problems with procrastination. |
| 8. Termination of activity 1 | General methods on how to cease activity, weekly report chart for termination of activity. |
| 9. Termination of activity 2 | Repetition and troubleshooting on activity initiation and termination, and goal-directed action. Behavior analysis of behavioral problems. |
| 10. Coping with stress 1 | General strategies and skills, weekly report chart to cope with stress. |
| 11. Coping with stress 2 | Behavior analysis of behavioral problems related to stress. |
| 12. Maintenance plan 1 | Maintaining behavioral changes after therapy |
| 13. Maintenance plan 2 | Managing setbacks. |
| 14. Maintenance plan 3 | Sticking with the maintenance plan. |

## Data collection and procedure

Recruitment started in June 2021 and went on until June 2022. It took place at three psychiatric outpatient centers representing areas with various socioeconomic status. Participants were treated in three separate groups, with group size ranging from 5 to 6 participants, due to restrictions on group size during the Covid-19 pandemic or limited base of recruitment at some locations. Groups were led by two clinical psychologists trained in the CADDI protocol and supervised by EES. In total, six group leaders were involved, including five licensed clinical psychologists and one resident psychologist. Individual interviews were scheduled immediately following treatment completion and conducted within three to five weeks, in July and August 2021 and in March, April, July and August 2022. A semi structured interview guide was developed by our research team through discussions regarding the study's objective and through two pilot interviews with individuals who had completed CADDI but were not part of the sample in this study. The interview guide included questions concerning participants' perceptions of CADDI. Initially some general questions were posed to start the conversation and to facilitate recall of the CADDI protocol, for example, "What was the overall aim of this treatment?", "What was the content of this treatment?" Thereafter more personal question regarding perceptions of treatment and its effect were asked; "How has the treatment affected you?", "What components of the treatment were most helpful to you?", Ultimately, questions regarding evaluation of treatment were asked; "Is there anything that could have made this treatment more helpful to you?", "Is there anything that was missing in treatment or that you would have liked to see more of? and "Did you experience any negative effects from treatment?". Interviews were performed by three licensed clinical CBT psychologists, familiar with adult ADHD, to understand treatment and thus participant responses. The interviews lasted 44,6 minutes on average (SD = 15.6, range = 22.2–75.0), and were conducted digitally via video sessions using a secure web platform. The interviewers were independent of the research group and this was explicitly stated by the interviewers before the interview started. The interviewers had no previous relationship with participants, to allow participants to express themselves freely, without consideration or perceived need to please the interviewer. The interviews were transcribed verbatim by assistants with little previous knowledge about CADDI and not involved in the analysis of data, to ensure transcriptions would not be influenced by preconceptions of the study objectives.

## Data analysis

We applied reflexive thematic analysis (rTA) and followed the outline of data analysis in six phases (familiarization with data, systematic coding of data, generating initial themes, developing and reviewing of themes, refining and defining themes, and writing the analysis) as described by Braun and Clarke [32–36], see Fig 1. In rTA and in several other qualitative methods, there are two approaches to analyzing data, inductive and deductive. An inductive approach refers to an analysis of data that is not guided by a specific theoretical framework, searching for patterns of meaning in the data, developing themes driven by the content in data. A deductive approach is guided by theory, and analysis is conducted through the lens of this theory, searching the data for information of relevance for theory [36]. The analysis in this study was primarily inductive, searching for patterns in the dataset, with no theoretical framework guiding the analysis, except for clinical and professional competence in the field. In the analysis of text, two epistemological perspectives can be used, an essentialist-realist and a constructivist where constructivism regards verbal statements as examples of underlying social constructions of meaning. We used an essentialist-realist perspective on language, considering verbal statements to be reflections of personal experience and meaning. We conducted the

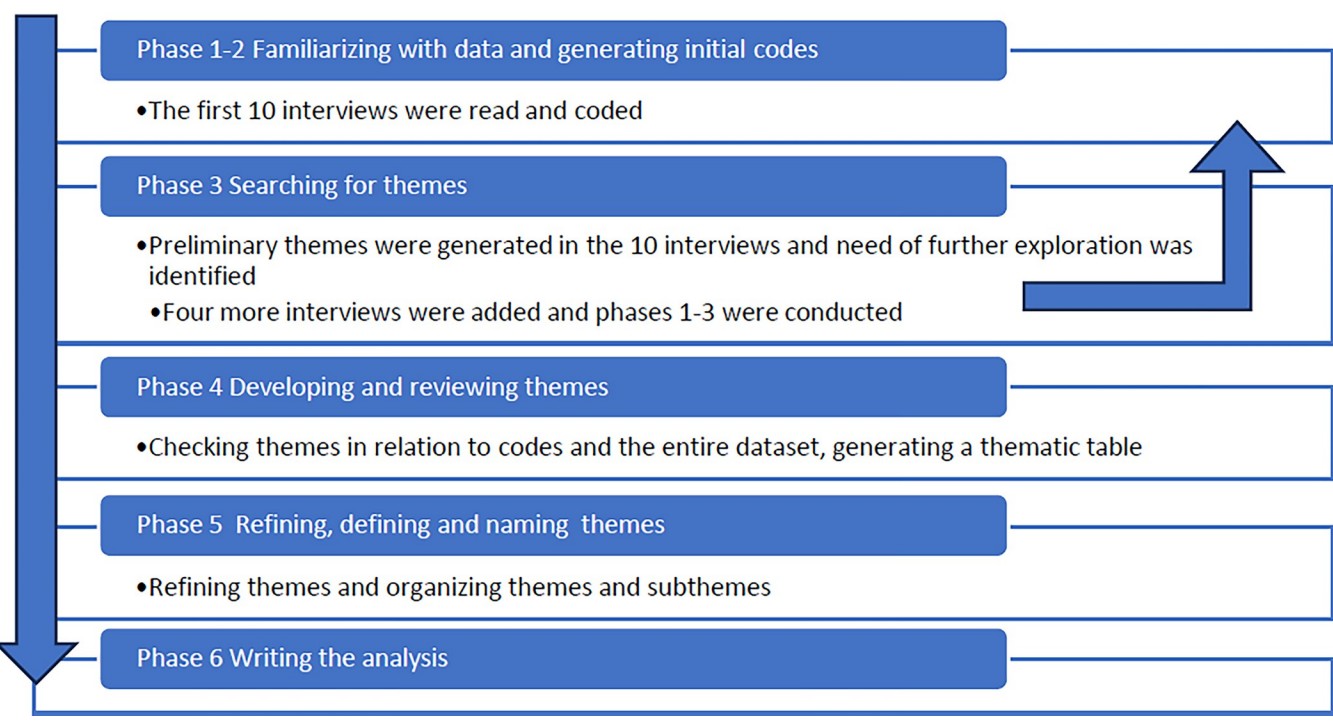

**Fig 1. The process of analyzing data in reflexive thematic analysis.**

analysis on a semantic and explicit level, (i.e., the meaning of verbal language as it manifests in verbal statements) not searching for latent meaning hidden between the lines. Sample size was determined by applying the concept of "pragmatic saturation", assuming that data collection is sufficient when the data give rich and multifaceted information regarding the research questions and new data does not contribute to new themes being generated [35]. Pragmatic saturation acknowledges the need for a substantial basis for analysis and represents the view that a completely exhaustive data collection and analysis can never be accomplished; thus data collection and analysis have to be terminated due to pragmatic circumstances (e.g., limited time and analytic resources) [33].

The data were analyzed by EES and RS in a collaborative and reflexive process, which was facilitated by BB, who structured the discussions. The analysis focused on broad thematic patterning throughout the dataset and was performed in two stages. Initially 10 of the 14 interviews were coded by EES and RS separately followed by a discussion of the analysis up to this point. These codes pointed to patterns of responses that could be labeled in preliminary themes and areas that needed further exploration. Subsequently, the four interviews from the last group were transcribed and the complete dataset was analyzed according to the phases described above. The last interviews enriched and deepened themes that had been developed in the first ten interviews and did not provide basis for new themes. The themes were then reviewed, named, and organized into main themes and subthemes. Altogether the dataset gave complex and multifaceted information regarding participants' perceptions of CADDI thereby the criteria of pragmatic saturation were met, as outlined in Braun and Clarke [35]. In total, three discussion meetings about codes, themes and mapping of themes were held throughout the analytic process. In rTA, researcher subjectivity and perspectives are considered as assets in the analytic process [36]. Thus, we choose researchers who brought different perspectives to

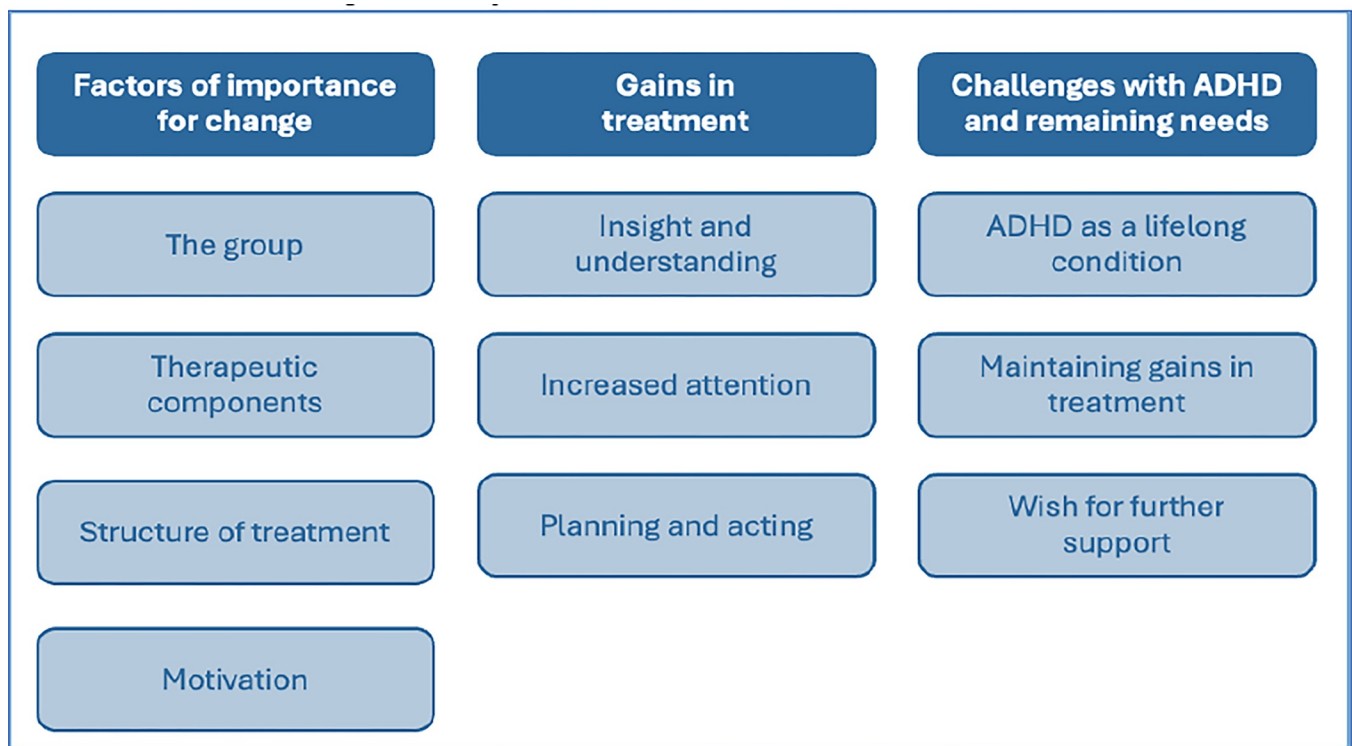

**Fig 2. Themes and subthemes generated from the dataset.**

the analysis; EES, MSc, is a licensed clinical psychologist and psychotherapist, specialized in CBT for ADHD. She has developed the CADDI protocol and is currently involved in conducting a multicenter trial of the protocol. In addition, EES has been involved as a group leader of CADDI and supervisor of group leaders in the groups participating in the RCT. RS, PhD, is a licensed clinical psychologist with no former knowledge of the CADDI protocol and no experience of CBT for ADHD. Thus, the different perspectives that EES and RS brought to the analysis, one involved and the other naïve, were considered beneficial to the analysis of data.

## Results

The analysis generated three themes (further described below and in Fig 2). The first theme was Factors of importance for change, including the subthemes; the group, therapeutic components, structure in treatment and motivation. The second theme regarding how treatment affected participants was called Gains in treatment, including the subthemes; insight and understanding, increased attention, planning and acting. Finally, a third theme called Challenges with ADHD-I and remaining needs was generated, including the subthemes; ADHD as a lifelong condition, maintaining gains in treatment and wish for further support.

### Factors of importance for change

Participants pointed to various factors contributing to change and personal growth during treatment. The group setting was perceived as vital, and the collaboration in the group was described as enhancing motivation and facilitating acceptance. Specific components of therapy and the structure of the treatment as well as motivation were also considered of importance for change.

**The group.** The group setting and the opportunity to meet others who had the same diagnosis was pointed out as a very important therapeutic factor that entailed several positive effects. Respondents described the group, where all participants shared the same predicaments and hardships, as a unique opportunity for sharing experiences and getting understood as well as accepted by one another. The group also brought an experience of belonging and being included in a supportive community:

*"Oftentimes you feel alone in things like these. You feel like nobody understands you, but coming here every Tuesday and feel like, these people understand me, we are here for one another, and we are moving in the same direction in a way. Yes, it was very important to me, I feel, because, oftentimes you feel that you are very much alone in things like these, and to meet others and feel that we are all in this together, then you kind of, then you get to understand each other, and you come closer to one another."* (Participant [P] 12)

Many participants expressed contentment with having a group exclusively for ADHD-I, in comparison to groups including participants with different diagnoses and various types of symptoms. Fellow participants with ADHD-I were appreciated for being calm and the narrow focus on ADHD-I was appreciated as it deepened the discussion of symptoms typical of ADHD-I. As succinctly put by one participant:

*"It was very nice to be in a room where everyone had ADD, that's kind of VIP"* (P7).

Through sharing and reflections of the group, the concept of the diagnosis and associated problem areas became clearer and more comprehensible. Sharing the same diagnosis also entailed trust and facilitated openness in the discussions of the group. Participants reported feeling more confident to receive help from each other when sharing the same predicament:

*"Somebody who has the same diagnosis, and the same problem knows exactly what you are talking about, and then you can embrace it in another way. And you can get it phrased in another way and that makes it easier to apprehend, I don't know, but I think that it's easier to embrace when you're likeminded and you don't have to feel that you're not like everybody else"* (P11).

**Therapeutic components.** Participants described the organizational skills taught in treatment as very helpful tools to increase everyday ability. Skills often mentioned in the interviews were using a calendar, creating routines, dividing big tasks into smaller parts, and using strategies to get started with tasks. Several participants also experienced mindfulness meditation to increase awareness and ability to stay focused on things of importance. With mindfulness it was easier to initiate activities and thus be more satisfied with oneself:

*"I've been able to use mindfulness in my everyday life and I feel that I get more done and I can be more content, also my self-confidence is strengthened as I see that I do things I promised to do"* (P10).

According to many participants, behavior analysis was a helpful tool to understand and illuminate difficulties, for instance, avoidance behaviors. Behavior analysis could also clarify consequences and help to choose actions more thoughtfully:

*"To analyze how to think in such situations, when, say, you've got off track, when you're in a slump, how to get out of it. Above all, behavior analysis has helped me a lot to see how to prioritize, or how to think in such situations"* (P3).

**Structure of treatment.**   Several participants raised the structure of treatment with weekly meetings and individual follow up through telephone calls as helpful. The home assignments included practicing new strategies between sessions and were followed up by group discussions; this practice was perceived as a supportive process in the acquisition of new skills. Recurrent reminders and the repetition of strategies and goals in treatment were also perceived as important to facilitate behavior change and cope with forgetting and being distracted:

*"Repeating long term goals and why you set up long term goals, as it is easy to forget why you set a goal, and why you are doing it. And repeating them constantly made you understand why you need to do it"* (P8).

**Motivation.**   Motivation and commitment in treatment was mentioned as important for adherence and work with home assignments. Motivation was talked about both as an attitude brought into treatment by the participants as well as something developed in treatment. Motivation was reported to be boosted by the structure and the group setting:

*"I also think that having a set time once a week does something to motivation and knowing that you are many and not alone in this, makes it easier to work towards your goals, and also to hear about the goal of the others and their daily struggle makes it easier and it doesn't feel as heavy on you then"* (P10).

Goal setting and how to move towards one's goals was perceived as a core aim of therapy that other components of treatment were intended to support. Participants described the importance of formulating for themselves why they wanted to do certain changes and strive towards specific goals to build motivation; as described by one participant:

*"to reflect on if this is something I really want or am I doing this to please someone else?"* (P7).

## Gains in treatment

Participants described effects of therapy in terms of gaining a long-awaited understanding of themselves which also entailed increased acceptance of their condition. Increased attention was also reported and a greater ability to direct one's behavior and to be true to one's intentions and goals.

**Insight and understanding.**   Many participants stated that they had gained knowledge of what the diagnosis of ADHD-I means and how it affects them in their everyday lives. Some participants reported being ignorant and confused about the impact of the diagnosis before treatment. During treatment, the insight of having an impairment became clearer. This ignorance of ADHD-I often resulted in keeping high standards of performance and not accepting the diagnosis as an impairment. Many participants reported that they often felt they were lazy and stupid and ought to perform as if they didn't have a neurodevelopmental disorder:

*"I've set expectations for myself to work as well as somebody who doesn't have ADD. So then, the expectations of myself are too high."* (P7)

Participants expressed the importance of understanding the diagnosis to know how to cope with their challenges and to realize what they could do to help themselves. Participants described that they got a clearer view of their problematic behaviors and what they could do to change them. Light was also shed on personal needs and how these needs could be met. Being

knowledgeable about their needs made it easier to care for themselves and set boundaries when necessary. Knowing themselves also made it easier to communicate with others:

*"When I understand myself better and can explain that to others, they understand me, and that have made a huge difference as I don't feel like I have to defend myself as I felt before, rather I can explain myself"* (P10).

Insight and understanding of their diagnosis and way of functioning paved the way for being more accepting and less judgmental, as reported by many participants. Increased self-tolerance was described as the result of a better understanding of symptoms and needs, and many respondents reported that a more accepting attitude was a major gain from therapy:

*So, I've got a little more, how to say, self-compassion. . . and . . . so I feel that I'm not as hard on myself as I used to be, and I think the possibility to talk to others going through the same things affected that pretty much"* (P8).

Finally, it was clear from the interviews that acceptance and being open to others are inter-related, for example, there were accounts of how talking to others about ADHD-I facilitated acceptance, and accounts of acceptance giving way to be more outspoken. With a more accepting attitude of one's difficulties it became easier to be open about them with others:

*"Yes, what's different in my everyday life is that I'm more open about having ADD. Or, I'm open about having difficulties with certain things and . . . then I can get help and then I notice that it gets easier for me and there are less misunderstandings. Instead of someone thinking that I'm lazy or ignorant it's the other way around; 'she really wants to, but she doesn't get it, I'll try another way.' So that makes a big difference. Thanks to the group treatment, I don't feel shame anymore for having it, as I have been among other fully functioning smart people who happen to have ADD"* (P10).

**Increased attention.**   Many participants described that they had become more attentive in their daily lives through the mindfulness meditation in therapy. There seemed to be an increased awareness of distractions and increased ability to return to one's intended focus. This made it easier to use strategies learned in treatment when needed, and less time was spent on distractions. Many participants described an increased ability to direct attention and thereby act in accordance with one's intention and values. Awareness and direction of attention were perceived as useful in many situations, for instance, when spending time with close ones:

*"I notice when I have strayed away in thoughts and I practice that [i.e., returning to the present] more often than before. In that way I have gotten better to stay present with others. . .and not just getting absent-minded."* (P2).

There was also an increased awareness of one's inner processes, such as thoughts and emotional states, as described in several accounts. Mindfulness, as taught in therapy, helped many participants to be more attentive to emotions and observe the role played by emotions in bothersome situations, as one participant reflected:

*"I'm more conscious of what I really feel. If I'm angry or irritated, I then say 'now I'm feeling this again' and then I'll take some time to do a mindfulness practice. And. . .yes.. so that's how*

*I've gotten more conscious of what I'm feeling and then I get less hard on myself, I don't think that I'm just lazy or bad"* (P13).

**Planning and acting.**   Many participants reported that they became more organized as they started to use strategies and spend time on planning their tasks. Developing routines for basic needs was mentioned as a helpful way to remember to take care of themselves and spend less effort on getting things done. There were many accounts of how starting to use a calendar and to-do list unburdened stress and increased contentment in everyday life as chores got done easier. Prioritizing according to one's long-term goals made it easier to move forward in the desired direction and strategies to get started were mentioned frequently as important to overcome tendencies to procrastinate:

*"I notice that I get a lot more done that are important to me due to treatment. These strategies are in my mind, for instance, we had a mindset we could use; to trick your brain to get started on things, by saying like, I'm only going to do this for five minutes, if it's something that feels really heavy"* (P10).

Many participants experienced difficulties initiating projects that they felt were huge and overwhelming, not knowing where to start. This changed when they learned to divide big tasks into smaller parts, set sub-targets, and take small steps on the way. There was a sense of competence in using planning skills wisely, and there were accounts of using skills in a pragmatic and flexible way to serve overall goals of wellbeing. Participants' planning skills also included being able to endure when plans don't work out and monitoring one's setbacks along the way. There seemed to be a more permissive attitude to a process of trial and error and to keep moving despite setbacks:

"*You plan much more, and then you fall off the horse, you do, and then you get up again"* (P9).

As evident in several interviews, making plans and acting in accordance with them gave an increased self-confidence. A stronger sense of self-respect came as a result of acting more in line with long-term goals.

## Challenges with ADHD-I and remaining needs

Participants were aware of the challenges with ADHD-I and that their symptoms would remain with them and needed to be cared for over time, as evident in the following sub-themes.

**ADHD as a lifelong condition.**   A majority of respondents talked about their challenges associated with ADHD-I as persistent symptoms that would affect their life long-term. Symptoms often mentioned were difficulties initiating activity, being forgetful and distractible. Many participants struggled with variability in performance level and a tendency to oscillate between extremes of activity and burnout. They described a challenge in finding balance in how to use one's energy and recover when weary:

*"When you've got energy, you feel that you're so creative and you should do a lot of stuff, and then you get stuck and just keep doing it, and then you are bedridden for 3 days because you've got no energy left and your brain can't operate because it's completely exhausted. You need to learn how to balance it better"* (P11).

Further, participants reporting the interventions in treatment to be helpful, such as mindfulness and organizational skills, were aware that they would need to maintain these practices to keep the benefits of them. However, the importance of acceptance and realistic demands were mentioned in many accounts as a means to cope with this lifelong challenge:

*"So, yeah, I still have to be very active and present, because it's still not like I have ceased to have ADD, so I must be aware of that. For some days, yeah, it's not the same everyday either, but some days I'm not that actionable and that. But also, with the insight that it's ok, I don't have to perform all the time."* (P7)

Many participants described what they had learned in treatment as something to keep striving for as an ongoing work in progress. For one participant, the importance of acceptance became clear during treatment and would entail continuous work from now on:

*"I think I needed and still need to work on acceptance of my diagnosis and not be so harsh on myself. . .because it's when I actually have accepted it, I can make something about it. Uhm, I've had to talk to myself a lot about accepting myself, and dare to ask for help; that's difficult. So those are goals I didn't write down and didn't have in therapy, but think is what I needed."* (P13)

**Maintaining gains in treatment.** Quite a few respondents discussed the issue of how they would maintain routines, strategies and acquired skills as the treatment group was terminated. Regular contact during treatment was perceived as very important to motivate effort to strive towards one's goals. There were two different attitudes represented in the interviews; one group of respondents was confident they would maintain their new habits, and the other group was more concerned about how to continue without the support from the group. Confidence in those who said they would stick to their maintenance plan and succeed seemed to stem from having made mindfulness practice and strategies to be organized and activated the new normal. There were also participants explaining how working to prepare for life after treatment in the maintenance plan had made it easier to see how to keep going and feeling confident:

*"Yeah, we made a maintenance plan and . . . it was kind of taking all that you had learned and perceived and writing it down in the maintenance plan. And then how to keep up the good work outside of treatment. . . It felt better, because I was kind of worried about how I would continue with everything after treatment. But after writing this plan it felt much easier, actually."* (P12)

Other participants who were less confident expressed their worries about how they would handle things on their own after the termination of the group. Some had already experienced difficulties sometime after the group's termination and described problems in keeping up with their strategies. Maintaining routines and skills on your own had turned out difficult:

*"I still find it hard to do this on my own, to use these strategies. It's like you´ve got the whole toolkit, but you still don't use it. I don't know. . ."* (P5).

**Wish for further support.** In some of the groups, participants started a group chat on social media to keep in touch with each other to provide mutual support after the treatment ended. A wish for follow-up meetings with the group emerged in interviews with both more

confident and more concerned participants. There was a longing to meet again and to continue a supportive community. There was also a wish to follow up on how everyone had gotten along after therapy terminated and to have an opportunity to recapitulate treatment and evaluate strategies after using them without the support of the group:

> *"To me it would have been helpful, to kind of, meet each other and talk again, kind of. If, say, somebody got, yeah, if somebody needs help or something. We do have the chat and that, but I think you could meet again in a while just to see how everyone is getting along. Yeah, because, you'd get to see how others might have dealt with setbacks and so on"* (P12).

## Discussion

The current study is one of few qualitative studies exploring participants perception of CBT for ADHD. The aim of the current study was to explore participants' perceptions of the treatment and what components were perceived as helpful regarding their needs. The study also sought to capture areas of improvement to make the protocol more useful to clients with ADHD-I. The analysis generated three themes: Factors of importance for change, Gains in treatment, and Challenges with ADHD-I and remaining needs. Participants were generally content with treatment, however, while admitting the persistent nature of their condition they also expressed concerns about the long-term effects of treatment.

Regarding factors of importance for change, the group setting was pointed out as vital. The group enhanced engagement in treatment and offered an open and tolerant atmosphere for sharing and reflecting with others, which facilitated acceptance. The experience of being one among many others with ADHD-I was described to ease the burden of shame, self-criticism, and loneliness, an effect that has been observed in previous studies of group interventions for ADHD [37,38]. The unambiguous appreciation of the group was consistently reported by interviewees representing all three groups and indicates particularly good group dynamics. Our data indicates the importance of homogeneity in symptoms among participants being vital for the positive perception of treatment in a group setting, and may influence therapeutic effect as previously observed in a meta- analysis of group psychotherapy [39]. Ambiguity regarding the impact of the group was reported in a study by Nordby and colleagues [38], where participants in "Goal management training" perceived the group as both supporting and distracting the work in the group. Heterogeneity regarding ADHD symptoms in the group caused feelings of exclusion in some participants thus pointing to challenges in group settings with various symptoms [38]. In our study the interviewees emphasized the importance of meeting and sharing their difficulties with others with ADHD-I and considered the exclusivity of ADHD-I in the group to be helpful to them. The appraisal of the group as a facilitator of self-awareness and acceptance supports the group setting as an integral part of the protocol.

A common statement concerned the value of gaining insight and understanding of ADHD-I through treatment. The many aspects of gaining knowledge about the diagnosis to understand oneself have been previously explored in a qualitative study by Hansson Halleröd and colleagues [40]. This study focused on the consequences of receiving a diagnosis of ADHD in adulthood and described that experience as multifaceted, affecting identity in several ways. The ADHD diagnosis brought an explanation for difficulties and entailed increased self-awareness and acceptance but also caused sorrow and disappointment over the lack of adequate support earlier in life [40]. In our study, all participants were diagnosed in adult age and had received a psychoeducational intervention regarding their diagnosis prior to treatment. Still, they underlined the value of understanding their diagnosis and its consequences

better through treatment. This increased insight was attributed to various elements in treatment; sharing experiences in the group, mindfulness, and behavior analysis all contributed to an increased understanding of their condition. Additionally, treatment in groups of individuals exclusively with ADHD-I, was perceived as deepening reflections and understanding of symptoms of inattention and associated difficulties.

Consistently, mindfulness was mentioned as a method to enhance attention and awareness of thoughts, feelings and activities and as a tool to facilitate goal directed behavior. This is in line with studies showing that mindfulness based cognitive therapy adapted for ADHD affected measures of both ADHD symptoms and executive functions [30,41]. In our study, participants confirmed that awareness of the present moment increased the ability to choose actions and remember to use strategies to deal with difficulties. This result provides support for the purpose of developing the CADDI protocol, which was to include both mindfulness and skills training regarding organization and activation in the same treatment [27]. Many participants also mentioned being more open and accepting of their emotions and more willing to see how emotions affected problem behavior. This result indicates feasibility of mindfulness in the treatment of inattention. It also supports the use of mindfulness practice in combination with CBT, in line with recommendations in a recent meta-analysis by Oliva et al. [42] where mindfulness-based interventions were found to be equally effective as other psychosocial interventions for ADHD.

In this study, we inquired about negative effects and asked for suggestions to make the treatment better. No negative effects were reported, which could be due to the small sample size and thus possible reporting bias. Further, there were no consistent suggestions regarding changes to the content of the protocol. The lack of response on how to improve the content could be due to previous work on developing and refining the protocol. Based on participants' responses in our feasibility study [27], revisions were made to improve CADDI regarding both content and delivery. The positive response to the protocol could be an effect of the administration, using both group and individual follow-up, thereby facilitating instructions and adjustments of the content to each individual. Moreover, the comparably small group size also might have been beneficial, allowing plenty of time for each participant and for discussions in the group. The homogeneity of symptoms also may have contributed to the appreciation of the content in the CADDI protocol. Further, the adjustments to ADHD-I in the protocol i.e., rehearsing themes and recurring follow-ups on all components of the protocol were described as helpful. Thereby the structure of the CADDI protocol, intended to support individuals with attention difficulties to benefit from treatment, were appreciated and seemed to serve their purpose.

Regarding the administration of the treatment, there were suggestions on how to make CADDI more useful. Quite a few participants were concerned about how to maintain gains in treatment by themselves over time, although the last three sessions were spent on making a maintenance plan. Participants were troubled by the risk of losing their routines due to the nature of their ADHD and suggested reunions of the group to continue supporting and learning from one another. The CADDI could be enhanced by adding booster sessions to support the maintenance of gains in treatment over time. Booster sessions are common within group settings in clinical care, but the effects of booster sessions are not certain as evaluated in a meta-analysis [43], pointing to possible positive effects while admitting lack of reporting and comparison groups in these studies. Previous studies of CBT for adult ADHD have used booster sessions following the intense phase of psychotherapy, thereby making the closing phase of therapy prolonged [15,44]. The impact of booster sessions in treatment of ADHD is still uncertain, one study comparing CBT in group with and without monthly booster sessions did not find significant difference between groups at follow up [15]. The authors concluded,

however, that some participants benefitted from booster sessions while others did not. Studies on the efficacy of booster sessions in CBT for ADHD are scarce, and this is an area in need of further attention.

## Strengths of the study

The sample was recruited from CADDI groups conducted in three different areas of Stockholm, representing various socio-economic conditions. Despite participants coming from separate groups led by various group leaders, participants' responses were consistent, pointing to widely shared perceptions of CADDI. These similarities increase the validity of results and support data saturation despite a relatively small sample size. Although the sample was small, it was similar to the population of completers in the CADDI condition in the RCT, regarding gender, age, and age when diagnosed, which strengthens the transferability of results. Interviews were conducted and transcribed by independent individuals, not involved in the research project, to avoid expectation bias from influencing the data collection.

## Limitations

In this study, we included a sample which was limited regarding cultural and clinical context, as the study was conducted in Swedish psychiatric outpatient centers and included participants who had sought psychological treatment and had been included in the RCT of CADDI. Further, we sampled completers and didn't invite individuals dropping out of treatment, who could have contributed some information on our research questions. Thus, the sample was small and therefore subjected to selection bias. Group size in this study was comparably small (5–6 participants) which could have contributed to the positive perception of the group and decreased the comparability of results with other studies of group interventions. Interviews were conducted within a larger timeframe than intended due to scheduling issues in the summertime, and this might have affected recall. However, memory of CADDI is hopefully kept since the protocol includes home assignments, is repetitive and lasts for 3,5 months.

## Implications for practice and further research

Our findings point to the value of conducting treatment in a presentation-specific format focusing exclusively on ADHD-I, as this was clearly described as a facilitator of certain therapeutic effects. To what extent homogeneity of symptoms is important for group dynamics and therapeutic effects is a question in need of continued exploration. Further, this study shows that participants felt that they would benefit more from an intervention that would support them over a more extended period, and the possible effect of a prolonged intervention ought to be subject of further studies (i.e., involving booster sessions). Supportive structures in the administration of CADDI (i.e., weekly telephone calls, practicing and repeating strategies and goals) were appreciated as helpful and, therefore, could be considered as possible adjustments in therapy with people suffering from executive dysfunction. The CADDI protocol has been evaluated in one open feasibility study [27] and the current qualitative study, both pointing to good feasibility and the value of a protocol designed specifically for ADHD-I. Further, the efficacy of the protocol is under investigation in an ongoing multicenter RCT comparing CADDI with treatment according to the Hesslinger protocol. While the CADDI protocol has been developed for adults, a focus for further research could be to adjust and test the protocol in adolescents with ADHD-I.

## Acknowledgments

The authors wish to express their gratitude to the participants and participating clinics Psychiatry Clinic Northwest within Stockholm Health Care Services, and Capio Psychiatry Jakobsberg Stockholm, Sweden, and to Elina Sebori, Sofia Asplund and Johanna Molin for conducting interviews.

## Author Contributions

**Conceptualization:** Elinor Eskilsson Strålin, Lisa B. Thorell, Tobias Lundgren, Sven Bölte, Benjamin Bohman.

**Data curation:** Elinor Eskilsson Strålin.

**Formal analysis:** Elinor Eskilsson Strålin, Rikard Sunnhed, Benjamin Bohman.

**Funding acquisition:** Elinor Eskilsson Strålin, Benjamin Bohman.

**Investigation:** Elinor Eskilsson Strålin, Benjamin Bohman.

**Methodology:** Elinor Eskilsson Strålin, Lisa B. Thorell, Tobias Lundgren, Sven Bölte, Benjamin Bohman.

**Supervision:** Benjamin Bohman.

**Writing – original draft:** Elinor Eskilsson Strålin.

**Writing – review & editing:** Rikard Sunnhed, Lisa B. Thorell, Tobias Lundgren, Sven Bölte, Benjamin Bohman.

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
