## [Decision Letter · Decision Letter 0]

2 Nov 2023

PONE-D-23-32070“It was very nice to be in a room where everyone had ADD - that’s kind of VIP” Clients’ Perspectives on Group CBT for ADHD Inattentive PresentationPLOS ONE

Dear Dr. Strålin,

Thank you for submitting your manuscript to PLOS ONE. After careful consideration, we feel that it has merit but does not fully meet PLOS ONE’s publication criteria as it currently stands. Therefore, we invite you to submit a revised version of the manuscript that addresses the points raised during the review process. Thank you for your submission, After careful review by several reviewers, we think the article requires major revisions prior to being accepted for publication. Please carefully review and address the concerns raised by reviewers, especially reviewer 9 who have recommended rejection. 

We look forward to receiving your revised manuscript.

Kind regards,

Lakshit Jain, MD

Academic Editor

PLOS ONE

Journal Requirements:

4. Thank you for stating the following financial disclosure: "The study was funded by the Mental Health Foundation and Professor Bror Gadelius Foundation. Open access funding was provided by Karolinska Institutet, Stockholm, Sweden. EES received a grant from Mental Health Foundation https://www.fondenpsykiskhalsa.se/?gclid=Cj0KCQjw1OmoBhDXARIsAAAYGSEWMemJkvxjxfpQjMhuoGWvHuQ_OIRp2E0wkClPx95TFxDbmMvfT3UaAsl7EALw_wcB

BB received a grant from Professor Bror Gadelius Foundation " ext-link-type="uri" xlink:type="simple">https://stiftelsemedel.se/stiftelsen-professor-bror-gadelius-minnesfond/".

Reviewers' comments:

Reviewer's Responses to Questions

**Comments to the Author**

1. Is the manuscript technically sound, and do the data support the conclusions?

Reviewer #1: Partly

Reviewer #2: Partly

Reviewer #3: Yes

Reviewer #4: Yes

Reviewer #5: Yes

Reviewer #6: Yes

Reviewer #7: Partly

Reviewer #8: Yes

Reviewer #9: Yes

2. Has the statistical analysis been performed appropriately and rigorously? 

Reviewer #1: Yes

Reviewer #2: N/A

Reviewer #3: Yes

Reviewer #4: I Don't Know

Reviewer #5: Yes

Reviewer #6: I Don't Know

Reviewer #7: N/A

Reviewer #8: N/A

Reviewer #9: N/A

3. Have the authors made all data underlying the findings in their manuscript fully available?

Reviewer #1: No

Reviewer #2: No

Reviewer #3: Yes

Reviewer #4: Yes

Reviewer #5: Yes

Reviewer #6: No

Reviewer #7: No

Reviewer #8: Yes

Reviewer #9: Yes

4. Is the manuscript presented in an intelligible fashion and written in standard English?

Reviewer #1: Yes

Reviewer #2: Yes

Reviewer #3: Yes

Reviewer #4: Yes

Reviewer #5: Yes

Reviewer #6: Yes

Reviewer #7: Yes

Reviewer #8: Yes

Reviewer #9: No

5. Review Comments to the Author

Reviewer #1: I have thoroughly reviewed your paper and appreciate your efforts. I've given constructive feedback to improve your article's quality. Please consider my suggestions to enhance clarity and impact. Once you make revisions, I will gladly review the updated version. Your commitment to improvement is commendable, and I look forward to seeing how your article evolves.

Best regards.

Abstract

1. The title is clear and descriptive. However, it could be made more concise by rephrasing it. For example, "Exploring the Experiences of Adults with ADHD-I in CADDI Treatment" could be a more succinct title.

2. Mention the duration of the interviews (e.g., the average duration of each interview) to provide a better understanding of the data collection process.

3. Specify the inclusion and exclusion criteria for participant selection, as this will help readers assess the study's external validity.

4. Consider providing more information on the demographics of the participants (e.g., age, gender, duration of ADHD-I diagnosis) to understand the diversity of the sample.

5. While the main themes and subthemes are clearly outlined, it would be helpful to provide a brief summary or key findings within the abstract to give readers a better sense of what the study discovered.

6. If there are any statistical or quantitative results, consider including key findings or statistics in the abstract.

7. The conclusions briefly summarize the main findings and emphasize the importance of the group, mindfulness, and follow-up sessions. However, it would be beneficial to provide more specific recommendations or implications for clinical practice or future research.

8. Ensure that the abstract adheres to the journal's specified word limit for abstracts. If it exceeds the limit, consider shortening it without sacrificing essential information.

9. Consider adding a sentence or phrase at the end of the abstract that highlights the broader significance of the study, its potential impact on clinical practice, or its contribution to the field of ADHD research.

Introduction

• The Introduction lacks a clear structure, making it challenging for readers to follow the argument. It would be helpful to organize the information in a more structured way. For instance, you can introduce the problem, briefly discuss existing treatments, and then state the need for the new CADDI protocol.

• For an enriched exploration of the ADHD (Attention-Deficit/Hyperactivity Disorder) subject, it is strongly recommended to draw upon the following authoritative sources:

1. https://brieflands.com/articles/ijhrba-82012.html

2. https://link.springer.com/chapter/10.1007/978-3-031-29368-9_18

3. https://www.ncbi.nlm.nih.gov/pmc/articles/PMC10124286/

4. https://brieflands.com/articles/ijpbs-108390.html#:~:text=Conclusions%3A,by%20prescribing%20proposed%20combined%20treatment.

• Use transition sentences or phrases to connect ideas and paragraphs. This will create a smoother flow of information and improve the readability of the text.

• While the text mentions the development of the CADDI protocol and its feasibility, it lacks a detailed rationale for why this new protocol is needed and how it addresses the gaps in existing treatments. Provide more context and justification for the CADDI protocol.

• It would be beneficial to explicitly state the hypotheses or objectives of the study in the Introduction. What specific questions or issues are you trying to address with this research? This will give readers a clear sense of what to expect.

Materials and Methods

It would be helpful to include a flowchart or diagram that visually represents the participant selection process, from recruitment to the final number of participants in the study. This can make it easier for readers to understand the process at a glance.

While you mention the concept of "information power," it would be beneficial to provide a brief justification for why the sample size chosen was adequate for the study. Explain how the sample size aligns with the study's goals and the qualitative research approach.

The section that discusses the interview guide is quite brief. It would be valuable to provide more information about the development of the guide, how it was refined after the pilot interviews, and how it was structured. This can help readers understand the quality of the data collection process.

In the section on data collection and procedure, you briefly mention that the interviewers were independent of the research group. Expound on the significance of this independence and how it was maintained throughout the study. This adds to the credibility of the data collection process.

While you provide a summary of the data analysis process, consider adding a visual representation (e.g., a flowchart or diagram) of the thematic analysis phases. This can help readers better comprehend the sequential steps involved in the analysis.

Mentioning that EES has developed the CADDI protocol and is currently involved in a multicenter trial on the protocol is important for transparency. However, it's essential to address the potential bias this may introduce. Discuss how potential biases were mitigated during the data analysis to maintain the study's objectivity.

In the data analysis section, you mention that the analysis was conducted on a "semantic level with an essentialist-realist perspective on language." Clarify what is meant by "semantic level" and "essentialist-realist perspective on language" to ensure clarity for readers who may not be familiar with these terms.

Results

1. After presenting the themes and subthemes, provide a brief summary or overview of each one. This will give readers a quick understanding of the main findings before delving into the details.

2. The inclusion of direct quotes from participants adds authenticity to the findings. Consider interspersing more quotes throughout the text to illustrate the themes and subthemes and make them more relatable.

3. If there are divergent or contrasting views within the themes, highlight them. Discuss how different participants' experiences may have influenced these perspectives. Acknowledging variations in responses can enrich the analysis.

4. Mention any potential biases or limitations that may have influenced the results, particularly in the context of the study's focus on the CADDI protocol. Discuss how the researchers attempted to mitigate biases during data collection and analysis.

5. Consider using visual aids such as tables, graphs, or charts to summarize the main themes and subthemes. Visual representations can help readers quickly grasp the key findings.

6. Examine whether the presentation adequately balances the coverage of each theme and subtheme. If one theme is more extensive than others, ensure that all themes receive a roughly equal level of detail and analysis.

Discussion

Begin the discussion by summarizing the primary aims of the study and a brief overview of the main findings. This provides readers with a clear context for the subsequent discussion.

When discussing the positive impact of the group setting, mindfulness, and other components of the CADDI protocol, consider comparing your findings to previous research. This can help contextualize your results and highlight the novelty of your study.

Acknowledge the limitations or areas of improvement suggested by the participants. Even though no negative effects were reported, discussing potential challenges or limitations can provide a more comprehensive view. For example, are there potential drawbacks or areas for future refinement of the protocol?

When discussing participants' concerns about maintaining gains in treatment, you mention the possibility of booster sessions. Elaborate on the concept of booster sessions, their potential effectiveness, and any research supporting this approach. Discuss whether this could be a useful addition to the CADDI protocol.

Conclude the discussion with a section on the implications of your findings for future research. Are there specific aspects of the CADDI protocol that require further investigation? Do you recommend more extensive studies to confirm the effectiveness of the CADDI protocol? Consider discussing research design and potential hypotheses for future studies.

Discuss the practical implications of your findings for clinicians or practitioners working with individuals with ADHD-I. How can your study inform the delivery of CBT for ADHD-I, and what practical recommendations can you provide?

As you discuss the implications of your findings, make it explicit what readers, whether they are researchers, clinicians, or policymakers, can take away from your study. What actionable insights can they apply in their work or further research?

While it's important to provide a thorough discussion, aim for conciseness by focusing on the most critical points. Avoid redundancy and repetition of points already covered in the results section.

Reviewer #2: Thank you for the opportunity to review this manuscript that describes a qualitative research study looking at the impacts of a short term group therapy treatment model for ADHD inattentive subtype.

Overall the manuscript was very well written. It was easy to read and had a good logical flow.

The qualitative findings including the three main themes of the factors for change, Treatment gains and challenges were discussed in good detail.

My main concern with the study is the lack of discussion around the limitations of such a study.

The sample size is small (14 patients) even for a qualitative study. There is no information on demographic factors other than gender. Even gender is skewed in non representative direction (9 females and only 5 males). This makes the generalizability of the results highly suspect.

This is highlighted by the responses around questions on negative effects. As the authors point out previous research has discussed challenges in a group setting (‘distracting the therapeutic work’). The fact that no participants were able to identify any negative effects, raises concerns for two things 1. Reporting bias and 2. Lack of heterogeneity in the sample. The discussion section ideally needs to acknowledge this and justify this better.

One other suggestion would be to make the discussion section more complete by placing the interpretations in context. How do the authors see these results as fitting in, in the clinical treatment of ADHD. How does the group therapy model potentially supplement medication treatment. A more nuanced discussion is needed.

There are a lots of merits to this kind of qualitative research. A revised manuscript could potentially address these concerns.

Reviewer #3: 1. I would like to thank you for the opportunity to review this article. The authors have done a great job in discussing CADDI( A group CBT protocol) as a treatment for ADHD-I. The ADHD-inattentive type is a very different entity from ADHD Hyperactivity/impulsivity type. There are differing views as to whether ADHD-I is a separate disorder or a subtype of ADHD.(1). The design of CADDI addressing unique features of ADHD-I which includes focusing on organizational skills, mindfulness in conjunction with behavioral analysis is a great strength of this study.

2. There is a need for more studies like these that address features unique to ADHD-I. There is a higher degree of prevalence of learning disabilities in this population. Inattention is a robust predictor of long term impairment. This subpopulation of ADHD is less likely to receive a correct diagnosis and treatment. The treatment for this condition must be multifaceted and a combination of different modalities to address the unique needs of this population must be employed.(2) This study is unique in addressing unique challenges of this condition in group CBT format.

3. However there are a few limitations of this study. The sample size of 14 is very small and participants were from Stockholm, Sweden. Perhaps this is a limitation of this study: small sample size and lack of diversity

SOLUTION: Please mention in limitation of the study that sample size was small. Also all participants were from Stockholm, Sweden. Perhaps the results would have been different if people from different cities in Sweden or from different countries in Europe were included.

4. The mean age of participants was 32.6 years. Also participants’ knew about their ADHD-I diagnosis which is not the case often. The mean age of presentation of ADHD-I is 12.4 years. A lot of adolescents drop out of high school resulting in limited possibilities for future employment. Usually these children drop out of school because of learning impairments.(2) So the study misses out on the group worst affected by it. Perhaps the age range could have been broader.

SOLUTION: Please include in limitation section that perhaps the study could have included wider age range to study the impact of CADDI on population of different age groups.

5. MINOR CORRECTIONS:

56: for health, social life and productivity, (that is) why it is paramount to find ways to decrease the

( THAT IS MISSING)

105: of July 2021 and vent (went) on until 31 of August 2022. Eligible participants in the trial were adults

( VENT WRONG SPELLING)

REFERENCES:

Grizenko N, Paci M, Joober R. Is the inattentive subtype of ADHD different from the combined/hyperactive subtype? J Atten Disord. 2010 May;13(6):649-57. doi: 10.1177/1087054709347200. Epub 2009 Sep 18. PMID: 19767592.

Weiss M, Worling D, Wasdell M. A chart review study of the inattentive and combined types of ADHD. J Atten Disord. 2003 Sep;7(1):1-9. doi: 10.1177/108705470300700101. PMID: 14738177.

Reviewer #4: 85 considered useful to deal with procrastination and passivity in clients with ADHD-I. REF PLEASE

88 would enhance the likelihood of behavior change. REF PLEASE

90 protocol specifically designed to address inattention and associated difficulties in ADHD-I. REF PLEASE

Methods: Limited sample size: The study only included 14 participants, which may not be representative of the broader population of individuals with ADHD-I.

Self-selection bias: Participants who chose to participate in the study may have been more motivated or interested in the topic, potentially skewing the results.

Recruitment location: Participants were recruited from psychiatric outpatient centers in Stockholm, which may not be representative of individuals with ADHD-I in other geographic regions or healthcare settings.

How authors addressed these above biases?

Data collection: Recall bias: Participants may have had difficulty accurately recalling their experiences with the CADDI treatment, especially if the interviews were conducted several weeks after treatment completion. I wonder how this was addressed?

To minimize these biases, researchers could consider using multiple methods to collect data, such as surveys or focus groups, to triangulate the data and improve the validity of the findings. Additionally, researchers could employ measures to minimize bias, such as using open-ended questions to avoid leading participants to certain responses, utilizing trained interviewers who are not involved in the treatment delivery, and conducting interviews closer to the completion of treatment to reduce recall bias

Overall, this study was conducted in a thorough and thoughtful manner, generating insightful results and stimulating discussion. The results provide valuable insights into the topic factors of Importance for Change, Progress in treatment and Challenges with ADHD-I and remaining needs. The discussion section provides a nuanced and thoughtful analysis of the implications of the findings, highlighting potential areas for future research and identifying practical applications. Overall, this study is a valuable contribution to the academic literature and provides a strong foundation for further inquiry.

Reviewer #5: A relevant and interesting topic is discussed. The study is aimed at evaluating group therapy participants experience of a new protocol used to treat attention deficit/hyperactivity disorder inattentive presentation (ADHD-I). The researchers utilized a new cognitive-behavioral therapy for ADHD-I (CADDI). Participants provided qualitative feedback about their experience. Within this feedback themes of participants experiences were identified including factors of importance for change, gains in treatment, and challenges with ADHD and remaining needs.

This paper provides additional research supporting the development of the CADDI. Although sample size was small, the addition of the concept of information of power provides support for the study. The authors also acknowledge that a completely exhaustive data collection is rare due to limited resources. However, in regards to the sample it may be beneficial to add other characteristics of the participants. For example, education level or IQ and age participant was diagnosed with ADHD are two possible confounds. These factors may affect a participants cognitive ability to participate in the CADDI protocol.

As for materials and methods, Table 1 provides a good visualization of the protocol. The methods section could benefit from a more detailed explanation of the timeline for interviews. Was there a reasoning behind waiting three to five weeks past treatment completion? Do you think the responses would have differed if the interviews occurred within a week of completion or longer than the five weeks?

• Line 105: spelling error “vent” should be “went”

• The inclusion criteria are addressed in lines 106 through 108. “Inclusion criteria included (i) stable use or no use of stimulations…” What is stable use of stimulants defined as? Do they have to be one the stimulants for a certain period of time?

. The research can be used to provide first hand knowledge of the participants experiences to future clients interested in the protocol. Clients often ask “why do group therapy?” or “how is this going to help me?” This research answers those common questions.

Treatment for ADHD seems to be moving toward a multimodal approach. Future research with this protocol could benefit from looking at the effect of prescribed medications along with therapy versus therapy only.

Reviewer #6: The manuscript is based on impressive empirical evidence and makes an original contribution, but findings cane be questioned because CADDI protocol itself is still in its initial stage of development and not yet proven standard effective method of therapy.

Reporting bias of the participants should also be commented and method to reduce it.

Results section should be shortened to improve the readability of the manuscript (may be eliminate the individual sentence response from participants as it doesn't require from my perspective).

Otherwise, its a very good and novel type of article and will be very interesting to see if this CADDI protocol becomes standard of CBT for ADD

Reviewer #7: PONE-D-23-32070

Thank you for the opportunity to review this manuscript “It was very nice to be in a room where everyone had ADD - that’s kind of VIP” Clients’. Perspectives on Group CBT for ADHD Inattentive Presentation.” The study aimed at capture 1) What aspects of treatment the participants found to be helpful, and 2) If there were areas that ought to be developed to make the protocol more useful to clients with ADHD-I. The authors designed a study with data collection using an interview protocol with semi-structured questions and invited individuals who had received treatment based on the CADDI protocol. The interview data was analysed with thematic analysis described by Braun and Clarke. It is a highly relevant topic and when designing therapeutic interventions, it´s always of value to have participants experiences.

1. Line 93

You saying that you aiming at explore participants’ perceptions of CADDI. In the

next sentence the concept: lived experience, is used. This is not in line with exploring perceptions with a semi-structured interview-guide. My suggestion: remove the sentence.

Also, the aim in the abstract, line 33,: …explored participants' experiences of … is not in line with the aim in the text, line 94, explore participants’ perceptions of…. Please use the same wording in the abstract and in the introduction and in the Discussion, line 455: explore participants’ perceptions.

2. Line 130

Participants were treated in three groups at separate locations, group-size ranged from 5 to 6 participants. What was the reasons for the relatively small groups, the group session was designed for 6-10 participants?

3. Line 150

Interviews were performed by three licensed clinical psychologists trained in CBT.

Why is their competence made clear? There is no explanation here or in the methodological considerations.

4. Line 166

The analysis was primarily inductive….

This is something that I can´t see in your result. Using questions such as: “How has the treatment affected you?”, “What components of the treatment were most helpful to you?”, “Is there anything that could have made this treatment more helpful to you?” and then end up with three themes: Factors of importance for change, Gains in treatment and

Challenges with ADHD-I and remaining needs.

Were these themes identified inductively? That´s difficult to see. When you have structured questions (to some degree) with specific questions in a relatively limited question area, it´s evidently very difficult to use inductive analysis. That may be that you used induction within the search for subthemes? Please rephrase this in the Data analysis.

5. Line 168

You are using “information power;” as a pragmatic saturation, and is referring to Braun Clarke. Please use the original author of this: Malterud K, Siersma VD, Guassora AD. Sample Size in Qualitative Interview Studies: Guided by Information Power. Qual Health Res. 2016 Nov;26(13):1753-1760. doi: 10.1177/1049732315617444. Epub 2016 Jul 10. PMID: 26613970.

6. Themes and subthemes

The analysis is surely well conducted and the subthemes seems to belong to their respective theme. But, naming the themes can be challenging and I don´t think that you have taking the time and discussions to do so. Now your themes more resemble purely grouping the statements. In describing the analysis, line 168, you are so nicely expressing: …considering verbal statements to be reflections of personal experience and meaning. Please change and/or develop naming of the themes, and some of the subthemes, so the result for us readers will be more descriptive and more in line with the participants experiences and meanings.

7. I don’t recognize any discussions of limitations or methodological considerations.

Please complete the Discussion with your thoughts of strengths and weaknesses.

Reviewer #8: Thank you for the opportunity to review this manuscript, which I enjoyed reading. It is well-written and describe thoroughly both methods used and the results from the qualitative interviews. It is important to listen to adults with ADHD on how they experience treatment interventions and not just exclusively use quantitative self-reports. I found the manuscript to be methodically of high quality. However, I am not an expert on the use of qualitative methods and may have overlooked methodological aspects of the manuscript that would be beneficial to improve.

I am one of the co-authors of the study of Nordby et al. (2021) and will as such make a note that this study was not a CBT but rather a neurocognitive training intervention in a group-setting. From reading the description of the CBT protocol used in the submitted study, I can clearly see that it is overlaps with the Goal Management Training (GMT) used in Nordby et al. But still GMT is not reckoned as a therapeutic intervention but rather a training intervention. This is an important distinction to make and relevant for the different findings discussed in the discussion section. It is interesting that across different types of interventions, adults with ADHD experience a group setting as beneficial and helpful by meeting other adults with similar difficulties. However, the training on executive functions in GMT may tap into the ADHD symptoms more compared with CBT by higher requirements on attentional control. This could be one of the reasons for why it was a link between heterogeneity in ADHD symptoms and feelings of exclusion in the Nordby study – such as that the training it-self will require a greater ability to be attentive and being less restless/fidgeting. As such, more severe ADHD symptoms can make the training on executive functions more challenging and give a feeling of not mastering the training as others in the same intervention group. I believe adding more on the distinction of the interventions and discuss differences in findings will enrich the discussion section and make it more relevant for the readers working clinically with or doing research on ADHD.

It is further very interesting that focusing on ADHD-I may have beneficial effects on the group setting of CBT. However, I think the authors could discuss the beneficial effects of focusing on ADHD subgroups against the general trend in intervention research to focus on transdiagnostic factors. It is not realistic in a clinical setting to have too many different interventions targeting specific mental health subgroups. This is one of the reasons for why there is an increased focus on developing and testing the effects of transdiagnostic interventions.

Reviewer #9: The article is wordy and can be edited to be more concise. Authors can consider formatting article in a clear sections, such as, limitation, and gaps in knowledge. Authors can consider elaborating if 14 candidates had 100% attendance; were any participants that was removed from study? The interviews were supervised by EES but it would be helpful to spell out that the structural interview was standardized.

Line 106-107 can authors elaborate "stable use." Also, it was unclear if participants were treated with non stimulant ADHD medication.

Line 108 authors can consider describing "severe mental illness."

6. PLOS authors have the option to publish the peer review history of their article (what does this mean?). If published, this will include your full peer review and any attached files.

Reviewer #1: No

Reviewer #2: No

Reviewer #3: **Yes: **JASLEEN KAUR MD

Reviewer #4: No

Reviewer #5: No

Reviewer #6: No

Reviewer #7: No

Reviewer #8: No

Reviewer #9: No

---

## [Author Response · Author response to Decision Letter 0]

15 Jan 2024

Response to reviewers 

We would like to express our appreciation for the comments from the reviewers, and the time and effort they have put into scrutinizing our manuscript. In the following, we list the reviewers’ comments and our response to each comment. Changes to the manuscript are highlighted using track changes.    

Reviewer #1: I have thoroughly reviewed your paper and appreciate your efforts. I've given constructive feedback to improve your article's quality. Please consider my suggestions to enhance clarity and impact. Once you make revisions, I will gladly review the updated version. Your commitment to improvement is commendable, and I look forward to seeing how your article evolves.

Best regards.

Abstract

1. The title is clear and descriptive. However, it could be made more concise by rephrasing it. For example, "Exploring the Experiences of Adults with ADHD-I in CADDI Treatment" could be a more succinct title. 

Authors’ response: Thank you for this suggestion. We have now changed the title to make it more clear that the study focuses on clients’ perspectives of the treatment. The title now reads as follows: “It was very nice to be in a room where everyone had ADD - that’s kind of VIP”: Exploring Clients’ Perceptions of Group CBT for ADHD Inattentive Presentation.

2. Mention the duration of the interviews (e.g., the average duration of each interview) to provide a better understanding of the data collection process.

Authors’ response: As requested, this information has been added in the abstract: “Individual semi-structured interviews, lasting on average 44,6 minutes, were conducted with 14 adults after the completion of CADDI”

3. Specify the inclusion and exclusion criteria for participant selection, as this will help readers assess the study's external validity.

Authors’ response: We have added this information in the method section (page 7), as there is little room for detail in the abstract. “Eligible participants in the RCT were adults with ADHD-I aged 18 years or older. Inclusion criteria were: (i) No change in medication two months prior to inclusion and (ii) completion of a psychoeducational course on ADHD focusing on symptoms, self-care and treatments. Exclusion criteria were: (i) severe mental illness (e.g., severe depression), (ii) substance abuse, or (iii) intellectual disability. Additional inclusion criteria for the current interview study were: (i) randomized to CADDI and (ii) completion (i.e. not dropping out) of the CADDI intervention.”

4. Consider providing more information on the demographics of the participants (e.g., age, gender, duration of ADHD-I diagnosis) to understand the diversity of the sample.

Authors’ response: We have added this information in the method section, as there is little room for detail in the abstract. Page 7 line 213-220 now reads: “Participants had a mean age of 32.6 years (SD=8.2, range= 22-50) and average age when first diagnosed with ADHD-I was 29.9 years (SD=9.1, range= 18-48). Nine (64%) participants were treated with stimulants with or without other pharmacotherapy, three (21 %) were medication free and two (14 %) were treated with non-stimulant medication. Regarding level of education, nine (64 %) participants had completed high school, four (29 %) had a bachelor’s or master’s degree and one (7 %) participant had not completed high school.”

5. While the main themes and subthemes are clearly outlined, it would be helpful to provide a brief summary or key findings within the abstract to give readers a better sense of what the study discovered.

Authors’ response: This is described under conclusions in the abstract which has been revised and now reads: “Participants emphasized the importance of the group setting as a facilitator of therapeutic effects of increased understanding and self-acceptance. Further, they considered the practice of mindfulness to enhance attention and awareness of thoughts, feelings, and activities and considered the structure of treatment as supporting the work in therapy. These findings support the value of the group setting and confirm the usefulness of CADDI. However, participants were concerned about how to maintain gains of treatment after its termination and suggested follow-up sessions to improve the CADDI protocol.”

6. If there are any statistical or quantitative results, consider including key findings or statistics in the abstract.

Authors’ response: This is a qualitative study, primarily searching for lived personal experiences in verbal statements. The only quantitative data included in this study is the sample description, which has now been added in the method section page 7. Page 7 line 213-220 now reads: “In the participant group, nine were females (64%) and mean age was 32.6 years (SD=8.2, range= 22-50) and average age when first diagnosed with ADHD-I was 29.9 years (SD=9.1, range= 18-48). Nine (64%) participants were treated with stimulants with or without other pharmacotherapy, three (21 %) were medication free and two (14%) were treated with non-stimulant medication. Regarding level of education, nine (64 %) participants had completed high school, four (29 %) had a bachelor’s or master’s degree and one (7 %) participant had not completed high school.”

7. The conclusions briefly summarize the main findings and emphasize the importance of the group, mindfulness, and follow-up sessions. However, it would be beneficial to provide more specific recommendations or implications for clinical practice or future research.

Authors’ response: Due to the strict word limit of the abstract, we have only been able to add some additional information. The conclusion section of the abstract now reads as follows: “Participants emphasized the importance of the group setting as a facilitator of therapeutic effects of increased understanding and self-acceptance. Further, they considered the practice of mindfulness to enhance attention and awareness of thoughts, feelings, and activities and considered the structure of treatment as supporting the work in therapy. These findings support the value of the group setting and confirm the usefulness of CADDI. However, participants were concerned about how to maintain gains of treatment after its termination and suggested follow-up sessions to improve the CADDI protocol.”

8. Ensure that the abstract adheres to the journal's specified word limit for abstracts. If it exceeds the limit, consider shortening it without sacrificing essential information. Consider adding a sentence or phrase at the end of the abstract that highlights the broader significance of the study, its potential impact on clinical practice, or its contribution to the field of ADHD research.

Authors’ response: We have revised the conclusion and kept it within the limit of 300 words.

9. The Introduction lacks a clear structure, making it challenging for readers to follow the argument. It would be helpful to organize the information in a more structured way. For instance, you can introduce the problem, briefly discuss existing treatments, and then state the need for the new CADDI protocol.

Authors’ response: We have revised the introduction to give a clearer structure, with paragraphs that discuss existing treatments and why a new treatment is needed, see pages 4-6.

10. For an enriched exploration of the ADHD (Attention-Deficit/Hyperactivity Disorder) subject, it is strongly recommended to draw upon the following authoritative sources:

1. https://brieflands.com/articles/ijhrba-82012.html

2. https://link.springer.com/chapter/10.1007/978-3-031-29368-9_18

3. https://www.ncbi.nlm.nih.gov/pmc/articles/PMC10124286/

4. https://brieflands.com/articles/ijpbs-108390.html#:~:text=Conclusions%3A,by%20prescribing%20proposed%20combined%20treatment.

Authors’ response: We have carefully considered the suggested references, but we find it difficult to grasp their potential added value of these studies to our work and the manuscript. The first study focuses on neurofeedback in adolescents and the third study on stimulant treatment in children. Thus, none of these studies focus on CBT or adults. Neither the second study (book chapter) nor the fourth study (study protocol) contains any original data. Therefore, we apologize and respectfully refrain from using these references as we do not think that they fit the purpose and study context. 

11. Use transition sentences or phrases to connect ideas and paragraphs. This will create a smoother flow of information and improve the readability of the text.

Authors’ response We have revised the text in the introduction, hopefully to be smoother and more readable, see pages 4-6.

12. While the text mentions the development of the CADDI protocol and its feasibility, it lacks a detailed rationale for why this new protocol is needed and how it addresses the gaps in existing treatments. Provide more context and justification for the CADDI protocol.

Authors’ response: The text has been revised to better explain the rationale for CADDI (page 5) which reads as follows: “Previous research has shown that treatment of inattention should preferably be comprehensive and multifaceted, including several components and strategies (20, 21). So far, no CBT protocol has been developed to address inattention and associated organizational difficulties using a broad range of interventions. To address the impairing consequences of inattention using CBT, it seems necessary to include components of both organizational skills, mindfulness and behavioral activation to deal with procrastination and passivity in clients with ADHD-I (25, 26). Based on these observations, the cognitive-behavioral therapy for ADHD-I (CADDI) protocol was developed.”

13.It would be beneficial to explicitly state the hypotheses or objectives of the study in the Introduction. What specific questions or issues are you trying to address with this research? This will give readers a clear sense of what to expect.

Authors’ response: The text has been revised to be clearer and more structured in the introduction (pages 4-6), as well as to better explain the rationale for CADDI (page 5). The objectives and research question have been re-worded to be clearer (page 6): “The objective of the current study was to explore participants’ perceptions of CADDI to better understand how participants experience the intervention and their opinions on how it might be further improved. Participants were asked to reflect on their perceptions of the treatment after completion. The exploration was guided by the following research questions; i) What aspects of treatment are helpful according to participants? and ii) Are there areas that ought to be developed to make the protocol more useful to clients with ADHD-I?” 

Materials and Methods

14. It would be helpful to include a flowchart or diagram that visually represents the participant selection process, from recruitment to the final number of participants in the study. This can make it easier for readers to understand the process at a glance.

Authors’ response: We have now made a clearer description of the participant selection process. This is explained in the manuscript see page 7: “Eligible participants in the RCT were adults with ADHD-I aged 18 years or older. Inclusion criteria were: (i) No change in medication two months prior to inclusion and (ii) completion of a psychoeducational course on ADHD focusing on symptoms, self-care and treatments. Exclusion criteria were: (i) severe mental illness (e.g., severe depression), (ii) substance abuse, or (iii) intellectual disability. Additional inclusion criteria for the current interview study were: (i) randomized to CADDI and (ii) completion (i.e. not dropping out) of the CADDI intervention. After completion of the CADDI protocol, all participants in the groups were informed about the study and invited to the interviews. All eligible 15 accepted to participate and gave their informed consent in writing.” After adding this information, we did not feel the need to include a flowchart.

15. While you mention the concept of "information power," it would be beneficial to provide a brief justification for why the sample size chosen was adequate for the study. Explain how the sample size aligns with the study's goals and the qualitative research approach. 

Authors’ response: This was explained in our manuscript and has been further elaborated on in the text. See page 12: “Sample size was determined by applying the concept of “pragmatic saturation”, assuming that data collection is sufficient when the data give rich and multifaceted information regarding the research questions and new data does not contribute to new themes being generated (35). Pragmatic saturation acknowledges the need for a substantial basis for analysis and represents the view that a completely exhaustive data collection and analysis can never be accomplished; thus, data collection and analysis have to be terminated du e to pragmatic circumstances (e.g., limited time and analytic resources) (33).”

We considered our sample of 14 interviews to be enough when both of these criteria were met. We analyzed ten interviews and found preliminary themes, but this was not saturated data. When analyzing the last four interviews, these merely enriched the initial themes from the first ten interviews but did not indicate additional ones. The 14 interviews were thematically consistent, which is why more interviews were not indicted according to pragmatic saturation

16. The section that discusses the interview guide is quite brief. It would be valuable to provide more information about the development of the guide, how it was refined after the pilot interviews, and how it was structured. This can help readers understand the quality of the data collection process.

Authors’ response: We have now described the development of the interview guide in more detail (page 10). “A semi-structured interview guide was developed by our research team through discussions of interview questions regarding the study’s objective and through two pilot interviews with individuals who had completed CADDI but were not part of the sample in this study. The interview guide included questions concerning participants’ perceptions of CADDI, initially some general questions were posed to start the conversation and to facilitate recall of the CADDI protocol, for example, “What was the overall aim of this treatment?”, “What was the content of this treatment?” Thereafter more personal question regarding perceptions of treatment and its effect were asked; “How has the treatment affected you?”, “What components of the treatment were most helpful to you?”, Ultimately, questions regarding evaluation of treatment were asked; “Is there anything that could have made this treatment more helpful to you?”, “Is there anything that was missing in treatment or that you would have liked to see more of? and “Did you experience any negative effects from treatment?”.

In the section on data collection and procedure, you briefly mention that the interviewers were independent of the research group. Expound on the significance of this independence and how it was maintained throughout the study. This adds to the credibility of the data collection process.

Authors’ response: Good point, we have added our view on the importance of independence of the interviewers and how we further handled this in the study. This now reads at page 11: “The interviewers were independent of the research group and this was explicitly stated by the interviewers before the interview started. The interviewers had no previous relationship with participants, to allow participants to express themselves freely, without consideration or perceived need to please the interviewer.”

18. While you provide a summary of the data analysis process, consider adding a visual representation (e.g., a flowchart or diagram) of the thematic analysis phases. This can help readers better comprehend the sequential steps involved in the analysis.

Authors’ response: The idea of a flow chart regarding the analytic process is very helpful and has been added to the manuscript at page 12.

Figure 1

19. Mentioning that EES has developed the CADDI protocol and is currently involved in a multicenter trial on the protocol is important for transparency. However, it's essential to address the potential bias this may introduce. Discuss how potential biases were mitigated during the data analysis to maintain the study's objectivity.

Authors’ response: When using reflexive thematic analysis (rTA) the concepts of objectivity and bias are regarded somewhat differently as compared to other qualitative approaches. We don’t necessarily seek to be “objective”, as the method of analyzing verbal statements could not be freed from the content of the minds doing the analysis. In rTA, knowledge in the field that is explored is regarded necessary to understand the data properly and is considered an asset. In this context transparency is important. Aware of EES’s and the other authors interest in the study, we invited a person not previously involved in our project to be the co-analyst of the data (RS). Page 14 now reads: “In rTA, researcher subjectivity and perspectives are considered as assets in the analytic process (36). Thus, we choose researchers who brought different perspectives to the analysis; EES, MSc, is a licensed clinical psychologist and psychotherapist, specialized in CBT for ADHD. She has developed the CADDI protocol and is currently involved in conducting a multicenter trial of the protocol. In addition, EES has been involved as a group leader of CADDI and supervisor of group leaders in the groups participating in the RCT. RS, PhD, is a licensed clinical psychologist with no former knowledge of the CADDI protocol and no experience of CBT for ADHD. Thus, the different perspectives that EES and RS brought to the analysis, one involved and the other naïve, were considered beneficial to the analysis of data.”

In the data analysis section, you mention that the analysis was conducted on a "semantic level with an essentialist-realist perspective on language." Clarify what is meant by "semantic level" and "essentialist-realist perspective on language" to ensure clarity for readers who may not be familiar with these terms.

Authors’ response: This is now described on page 12 “In the analysis of text, two epistemological perspectives can be used, an essentialist-realist and a constructivist where constructivism regards verbal statements as examples of underlying social constructions of meaning. We used an essentialist-realist perspective on language, considering verbal statements to be reflections of personal experience and meaning. We conducted the analysis on a semantic and explicit level, (i.e., the meaning of verbal language as it manifests in verbal statements) not searching for latent meaning hidden between the lines.” 

Results

1. After presenting the themes and subthemes, provide a brief summary or overview of each one. This will give readers a quick understanding of the main findings before delving into the details.

Authors’ response: We have revised the first paragraph of the results. See page 14

“ The analysis generated three themes (further described below and in Figure 2). The first theme was Factors of importance for change, including the subthemes; the group, therapeutic components, structure in treatment and motivation. The second theme regarding how treatment affected participants was called Gains in treatment, including the subthemes; insight and understanding, increased attention, planning and acting. Finally, a third theme called Challenges with ADHD-I and remaining needs was generated, including the subthemes; ADHD as a lifelong condition, maintaining gains in treatment and wish for further support.” 

2. The inclusion of direct quotes from participants adds authenticity to the findings. Consider interspersing more quotes throughout the text to illustrate the themes and subthemes and make them more relatable.

Authors’ response: We like the quotes as well, but we have also been requested by other reviewers to write more concise and remove redundant quotes so at this point we choose to not add more quotes.

3. If there are divergent or contrasting views within the themes, highlight them. Discuss how different participants' experiences may have influenced these perspectives. Acknowledging variations in responses can enrich the analysis.

Authors’ response: In this study we used reflexive thematic analysis, and we describe themes that are generated in the dataset, and generally there were highly corresponding views stated in our data. One of our themes are made of divergent opinions around one central topic, see for example “maintaining gains in treatment” described at page 24. 

4. Mention any potential biases or limitations that may have influenced the results, particularly in the context of the study's focus on the CADDI protocol. Discuss how the researchers attempted to mitigate biases during data collection and analysis.

Authors response: We have elaborated our thoughts on this under “Limitations” se page 31-32 “In this study, we included a sample which was limited regarding cultural and clinical context, as the study was conducted in Swedish psychiatric outpatient centers and included participants who had sought psychological treatment and had been included in the RCT of CADDI. Further, we sampled completers and didn’t invite individuals dropping out of treatment, who could have contributed some information on our research questions. Thus, the sample was small and therefore subjected to selection bias.”

5. Consider using visual aids such as tables, graphs, or charts to summarize the main themes and subthemes. Visual representations can help readers quickly grasp the key findings.

Authors’ response: We had originally one table to visualize our themes and subthemes and now we have turned it into a figure to visualize our data. 

6. Examine whether the presentation adequately balances the coverage of each theme and subtheme. If one theme is more extensive than others, ensure that all themes receive a roughly equal level of detail and analysis.

Authors’ response: Thank you for these valuable comments on the results, the themes vary in complexity and coverage, to us it’s important that all themes are well described and illuminated by adequate quotes. Upon re-examination of the presentation of themes and subthemes, we believe that the presentation is adequate. While all materials were subjected to an equal level of analysis, some themes and subthemes were more prominent or complex than others; thus, we believe it is natural that the coverage varies across themes and subthemes.

Discussion

1. Begin the discussion by summarizing the primary aims of the study and a brief overview of the main findings. This provides readers with a clear context for the subsequent discussion.

Authors’ response Thank you for this suggestion. The first paragraph of the Discussion now reads: “The aim of the current study was to explore participants’ perceptions of the treatment and what components were perceived as helpful regarding their needs. The study also sought to capture areas of improvement to make the protocol more useful to clients with ADHD-I. The analysis generated three themes: Factors of importance for change, Gains in treatment, and Challenges with ADHD-I and remaining needs. Participants were generally content with treatment, however, while admitting the persistent nature of their condition they also expressed concerns about the long-term effects of treatment.” 

2. When discussing the positive impact of the group setting, mindfulness, and other components of the CADDI protocol, consider comparing your findings to previous research. This can help contextualize your results and highlight the novelty of your study.

Authors’ response: We have rewritten the Discussion according to your suggestions. Please see pages For example in this section, page 27-28: “Our data indicates the importance of homogeneity in symptoms among participants being vital for the positive perception of treatment in a group setting, and may influence therapeutic effect as previously observed in a meta- analysis of group psychotherapy (39). Ambiguity regarding the impact of the group was reported in a study by Nordby and colleagues (38), where participants in “Goal management training” perceived the group as both supporting and distracting the work in the group.”

3. Acknowledge the limitations or areas of improvement suggested by the participants. Even though no negative effects were reported, discussing potential challenges or limitations can provide a more comprehensive view. For example, are there potential drawbacks or areas for future refinement of the protocol?

Authors’ response: Thank you for these valuable suggestions for our discussion! We have revised the discussion and inserted the sections Strengths of the study, Limitations and Implications for practice and further research. Please see pages 27-33.

4. When discussing participants’ concerns about maintaining gains in treatment, you mention the possibility of booster sessions. Elaborate on the concept of booster sessions, their potential effectiveness, and any research supporting this approach. Discuss whether this could be a useful addition to the CADDI protocol.

Authors’ response: Thank you for these valuable suggestions for our discussion. We have elaborated the discussion and our thoughts regarding booster sessions. Please see page 31 “Participants were troubled by the risk of losing their routines due to the nature of their ADHD and suggested reunions of the group to continue supporting and learning from one another. The CADDI could be enhanced by adding booster sessions to support the maintenance of gains in treatment over time. Booster sessions are common within group settings in clinical care, but the effects of booster sessions are not certain as evaluated in a meta-analysis (43), pointing to possible positive effects while admitting lack of comparison groups in these studies. Previous studies of CBT for adult ADHD have used booster sessions following the intense phase of psychotherapy, thereby making the closing phase of therapy prolonged (15, 44). The impact of booster sessions in treatment of ADHD is still uncertain, one study comparing CBT in group with and without monthly booster sessions did not find significant difference between groups at follow up (15). The authors concluded, however, that some participants benefitted from booster sessions while others did not. Studies on the efficacy of booster sessions in CBT for ADHD are scarce, and this is an area in need of further attention. “

Conclude the discussion with a section on the implications of your findings for future research. Are there specific aspects of the CADDI protocol that require further investigation? Do you recommend more extensive studies to confirm the effectiveness of the CADDI protocol? Consider discussing research design and potential hypotheses for future studies.

Authors’ response: Thank you. We have revised the discussion and inserted the section Implications for practice and further research where we give a few suggestions regarding future studies. Please page 32-33.

Discuss the practical implications of your findings for clinicians or practitioners working with individuals with ADHD-I. How can your study inform the delivery of CBT for ADHD-I, and what practical recommendations can you provide?

As you discuss the implications of your findings, make it explicit what readers, whether they are researchers, clinicians, or policymakers, can take away from your study. What actionable insights can they apply in their work or further research?

While it’s important to provide a thorough discussion, aim for conciseness by focusing on the most critical points. Avoid redundancy and repetition of points already covered in the results section.

Authors’ response: Thank you for these valuable suggestions for our discussion. We have added sections and revised the discussion and elaborated on our thoughts regarding implication for practice, for example on importance of homogeneity in groups and adapting treatments to individuals with executive dysfunction. Se page 31-33. The section Implications for practice and further research now reads: ”Our findings point to the value of conducting treatment in a presentation-specific format focusing exclusively on ADHD-I, as this was clearly described as a facilitator of certain therapeutic effects. To what extent homogeneity of symptoms is important for group dynamics and therapeutic effects is a question in need of continued exploration. Further, this study shows that participants felt that they would benefit more from an intervention that would support them over a more extended period, and the possible effect of a prolonged intervention ought to be subject of further studies (i.e., involving booster sessions). Supportive structures in the administration of CADDI (i.e., weekly telephone calls, practicing and repeating strategies and goals) were appreciated as helpful and, therefore, could be considered as possible adjustments in therapy with people suffering from executive dysfunction. The CADDI protocol has been evaluated in one open feasibility study (27) and the current qualitative study, both pointing to good feasibility and the value of a protocol designed specifically for ADHD-I. Further, the efficacy of the protocol is under investigation in an ongoing multicenter RCT comparing CADDI with treatment according to the Hesslinger protocol. While CADDI protocol has been developed for adults, a focus for further research could be to adjust and test the protocol in adolescents with ADHD-I. “

Reviewer #2: Thank you for the opportunity to review this manuscript that describes a qualitative research study looking at the impacts of a short term group therapy treatment model for ADHD inattentive subtype. Overall the manuscript was very well written. It was easy to read and had a good logical flow. The qualitative findings including the three main themes of the factors for change, Treatment gains and challenges were discussed in good detail. My main concern with the study is the lack of discussion around the limitations of such a study. 

1. The sample size is small (14 patients) even for a qualitative study. 

Authors’ response: Sample size is decided using pragmatic saturation. As discussed above (see comment #15 reviewer 1, page 5 in this document), a sample size of 14 patients was considered enough considering the fact that no new themes were identified after the first 10 interviews. Thus, saturation was clearly reached with a sample size of 14. We have elaborated on this in our manuscript. See page 12: “Sample size was determined by applying the concept of “pragmatic saturation”, assuming that data collection is sufficient when the data give rich and multifaceted information regarding the research questions and new data does not contribute to new themes being generated (35). Pragmatic saturation acknowledges the need for a substantial basis for analysis and represents the view that a completely exhaustive data collection and analysis can never be accomplished; thus, data collection and analysis have to be terminated due to pragmatic circumstances (e.g., limited time and analytic resources) (33).”

2. There is no information on demographic factors other than gender. Even gender is skewed in non representative direction (9 females and only 5 males). This makes the generalizability of the results highly suspect.

Authors’ response: Thanks for this comment. We have added more information on demographic factors both of our sample and of the population of interest, which shows that the sample and the population are similar. As further discussed below in response to the next comment, the skewed male-female ratio is a result of the fact that more females than males generally seek psychological treatment for ADHD and this was the case also in our study. We have added this information to the manuscript, see page 7: “Participants had a mean age of 32.6 years (SD=8.2, range= 22-50) and average age when first diagnosed with ADHD-I was 29.9 years (SD=9.1, range= 18-48). Nine (64%) participants were treated with stimulants with or without other pharmacotherapy, three (21 %) were medication free and two (14 %) were treated with non-stimulant medication. Regarding level of education, nine (64 %) participants had completed high school, four (29 %) had a bachelor’s or master’s degree and one (7 %) participant had not completed high school. In the RCT there was 53 completers of CADDI, of them 34 (64.2%) were females, they had a mean age of 35.3 (SD=8.4, range 21-53) years, and mean age when first diagnosed was 31.6 years (SD=9.5 range 12-52).”

3. This is highlighted by the responses around questions on negative effects. As the authors point out previous research has discussed challenges in a group setting (‘distracting the therapeutic work’). The fact that no participants were able to identify any negative effects, raises concerns for two things 1. Reporting bias and 2. Lack of heterogeneity in the sample. The discussion section ideally needs to acknowledge this and justify this better.

Authors’ response: Thank you for these valuable questions. 1. Regarding reporting bias, we handled this through inviting everyone that completed the treatment in three groups (15 individuals), everyone volunteered to be interviewed, however one participant canceled the interview, due to difficulties scheduling the interview. Further we used independent interviewers to avoid expectation bias to influence data collection. However, as this is a selected sample and the sample size is small, selection bias could not be ruled out. 2. As the focus is perceptions of CBT treatment, it is important that the sample is representative of ADHD-I adults seeking psychological treatment. We have revised the manuscript and elaborated on our thoughts regarding these questions in the Discussion page 27-33 and under “Limitations”, see page 31-32, which now reads: Limitations “In this study, we included a sample which was limited regarding cultural and clinical context, as the study was conducted in Swedish psychiatric outpatient centers and included participants who had sought psychological treatment and had been included in the RCT of CADDI. Further, we sampled completers and didn’t invite individuals dropping out of treatment, who could have contributed some information on our research questions. Thus, the sample was small and therefore subjected to selection bias. Group size in this study was comparably small (5-6 participants) which could have contributed to the positive perception of the group and decreased the comparability of results with other studies of group interventions. Interviews were conducted within a larger timeframe than intended due to scheduling issues in the summertime, and this might have affected recall. However, memory of CADDI is hopefully kept since the protocol includes home assignments, is repetitive and lasts for 3,5 months.” 

4. One other suggestion would be to make the discussion section more complete by placing the interpretations in context. How do the authors see these results as fitting in, in the clinical treatment of ADHD. How does the group therapy model potentially supplement medication treatment. A more nuanced discussion is needed.

Authors’ response: Thank you, we have developed the discussion (pages 27-33) with subheadings like, “Strengths of the study”, “Limitations” and “Implications for practice and future research where we further elaborate our thought on these questions, page 32 “Implications for practice and further research Our findings point to the value of conducting treatment in a presentation-specific format focusing exclusively on ADHD-I, as this was clearly described as a facilitator of certain therapeutic effects. To what extent homogeneity of symptoms is important for group dynamics and therapeutic effects is a question in need of continued exploration. Further, this study shows that participants felt that they would benefit more from an intervention that would support them over a more extended period, and the possible effect of a prolonged intervention ought to be subject of further studies (i.e., involving booster sessions). Supportive structures in the administration of CADDI (i.e., weekly telephone calls, practicing and repeating strategies and goals) were appreciated as helpful and, therefore, could be considered as possible adjustments in therapy with people suffering from executive dysfunction. The CADDI protocol has been evaluated in one open feasibility study (27) and the current qualitative study, both pointing to good feasibility and the value of a protocol designed specifically for ADHD-I. Further, the efficacy of the protocol is under investigation in an ongoing multicenter RCT comparing CADDI with treatment according to the Hesslinger protocol. While CADDI protocol has been developed for adults, a focus for further research could be to adjust and test the protocol in adolescents with ADHD-I.” 

5. There are a lots of merits to this kind of qualitative research. A revised manuscript could potentially address these concerns.

Authors’ response: Thank you!

Reviewer #3: 1. I would like to thank you for the opportunity to review this article. The authors have done a great job in discussing CADDI( A group CBT protocol) as a treatment for ADHD-I. The ADHD-inattentive type is a very different entity from ADHD Hyperactivity/impulsivity type. There are differing views as to whether ADHD-I is a separate disorder or a subtype of ADHD.(1). The design of CADDI addressing unique features of ADHD-I which includes focusing on organizational skills, mindfulness in conjunction with behavioral analysis is a great strength of this study.

1. There is a need for more studies like these that address features unique to ADHD-I. There is a higher degree of prevalence of learning disabilities in this population. Inattention is a robust predictor of long term impairment. This subpopulation of ADHD is less likely to receive a correct diagnosis and treatment. The treatment for this condition must be multifaceted and a combination of different modalities to address the unique needs of this population must be employed. (2) This study is unique in addressing unique challenges of this condition in group CBT format.

Authors’ response: Thank you for providing this information and useful references, they have been inserted in the introduction on page 5 in the manuscript.

2. However there are a few limitations of this study. The sample size of 14 is very small and participants were from Stockholm, Sweden. Perhaps this is a limitation of this study: small sample size and lack of diversity

SOLUTION: Please mention in limitation of the study that sample size was small. Also all participants were from Stockholm, Sweden. Perhaps the results would have been different if people from different cities in Sweden or from different countries in Europe were included.

Authors´ response: Thank you for these valuable comments. We have revised the manuscript and elaborated on our thoughts regarding the sample under “Limitations” see page 31. It is a selected sample in this study and further research in other countries would be highly relevant. Regarding sample size, we used pragmatic saturation and to decide on sample size. This has been elaborated on in the manuscript to be clearer. Please see page 12 “Sample size was determined by applying the concept of “pragmatic saturation”, assuming that data collection is sufficient when the data give rich and multifaceted information regarding the research questions and new data does not contribute to new themes being generated (35). Pragmatic saturation acknowledges the need for a substantial basis for analysis and represents the view that a completely exhaustive data collection and analysis can never be accomplished; thus, data collection and analysis have to be terminated due to pragmatic circumstances (e.g., limited time and analytic resources) (33).”

3. The mean age of participants was 32.6 years. Also participants’ knew about their ADHD-I diagnosis which is not the case often. The mean age of presentation of ADHD-I is 12.4 years. A lot of adolescents drop out of high school resulting in limited possibilities for future employment. Usually these children drop out of school because of learning impairments.(2) So the study misses out on the group worst affected by it. Perhaps the age range could have been broader. 

SOLUTION: Please include in limitation section that perhaps the study could have included wider age range to study the impact of CADDI on population of different age groups.

Authors’ response: Thank you, we have provided more information on demographic data and age of first being diagnosed in the text. See page 7 “Participants had a mean age of 32.6 years (SD=8.2, range= 22-50) and average age when first diagnosed with ADHD-I was 29.9 years (SD=9.1, range= 18-48).” However, we do not believe that it is a limitation of the present study that it concerned adults, as the CADDI protocol was designed only for adults. However, it would be very interesting to investigate the effects of CADDI in adolescents, which we now have included as a possible area for future research (page 32-33 which now reads: While the CADDI protocol has been developed for adults, a focus for further research could be to adjust and test the protocol in adolescents with ADHD-I.” 

5. MINOR CORRECTIONS:

56: for health, social life and productivity, (that is) why it is paramount to find ways to decrease the

( THAT IS MISSING)

105: of July 2021 and vent (went) on until 31 of August 2022. Eligible participants in the trial were adults

( VENT WRONG SPELLING) 

Authors response: Thank you for these observations. All spelling errors have now been corrected (pages 4 and 9).

REFERENCES:

Grizenko N, Paci M, Joober R. Is the inattentive subtype of ADHD different from the combined/hyperactive subtype? J Atten Disord. 2010 May;13(6):649-57. doi: 10.1177/1087054709347200. Epub 2009 Sep 18. PMID: 19767592.

Weiss M, Worling D, Wasdell M. A chart review study of the inattentive and combined types of ADHD. J Atten Disord. 2003 Sep;7(1):1-9. doi: 10.1177/108705470300700101. PMID: 14738177.

Reviewer #4: 85 considered useful to deal with procrastination and passivity in clients with ADHD-I. REF PLEASE 

88 would enhance the likelihood of behavior change. REF PLEASE

90 protocol specifically designed to address inattention and associated difficulties in ADHD-I. REF PLEASE

Authors’ response: Thank you for pointing to the need of references, we have added references to the manuscript, as suggested. 

2. Methods: Limited sample size: The study only included 14 participants, which may not be representative of the broader population of individuals with ADHD-I.

Authors´ response: Thank you for these valuable comments. In this study, our objective was to explore the perception of CADDI in those participants that completed treatment and therefore we purposively selected this sample of individuals, meeting our inclusion criteria, see pages 6-7. The 14 participants were sampled from a group of 53 eligible individuals in the RCT, that is completers in the CADDI condition. To us it is important that the sample is representative of the larger group of participants in the RCT study and therefore we have inserted information regarding this in our manuscript at page 7 which now reads; “In the RCT there was 53 completers of CADDI, of them 34 (64.%) were females, they had a mean age of 35.3 (SD=8.4, range 21-53) years, and mean age when first diagnosed was 31.6 years (SD=9.5 range 12-52).”

Regarding sample size for our objective, we used pragmatic saturation to decide on sample size as described in a previous comment see reviewer #3 #3. We have revised the manuscript and elaborated on our thoughts regarding the sample under “Limitations”. See page 31.

3. Self-selection bias: Participants who chose to participate in the study may have been more motivated or interested in the topic, potentially skewing the results.

Author’s response: We invited 15 people, that is everyone who completed treatment in the three groups, and all 15 volunteered to be interviewed. However, 14 of them managed to schedule an interview although in the middle of the holiday season. This information has been added (page 7). “After completion of the CADDI protocol, all participants in three groups at separate locations, were informed about the study and invited to the interviews. All 15 accepted to participate and gave their informed consent in writing. Fourteen of them managed to schedule an interview for the study.”

4. Recruitment location: Participants were recruited from psychiatric outpatient centers in Stockholm, which may not be representative of individuals with ADHD-I in other geographic regions or healthcare settings. How authors addressed these above biases?

Authors’ response: We have revised the manuscript and elaborated on our thoughts regarding the sample’s representativity under “Limitations” see page 31 In this study, we included a sample which was limited regarding cultural and clinical context, as the study was conducted in Swedish psychiatric outpatient centers and included participants who had sought psychological treatment and had been included in the RCT of CADDI. Further, we sampled completers and didn’t invite individuals dropping out of treatment, who could have contributed some information on our research questions. Thus, the sample was small and therefore subjected to selection bias.”Further, we have clarified the text regarding inclusion in the study. It is a selected sample, but in regard to our objective, these participants are purposively sampled.

5. Data collection: Recall bias: Participants may have had difficulty accurately recalling their experiences with the CADDI treatment, especially if the interviews were conducted several weeks after treatment completion. I wonder how this was addressed?

Author’s response: Thanks for this comment. We aimed at conducting interviews as soon as possible after completing CADDI treatment. Unfortunately, scheduling difficulties in both interviewee and interviewers hindered this ambition. Some groups ended close to the summer season with vacation and traveling among the interviewees affecting scheduling of interviews. Thus, interviews were performed within a larger timeframe than we had intended. To recall the CADDI, the interviews started with general, open questions regarding the aim and components of CADDI. We have revised the text in the discussion regarding this se under Limitations page 31 “Interviews were conducted within a larger timeframe than intended due to scheduling issues in the summertime, and this might have affected recall. However, memory of CADDI is hopefully kept since the protocol includes home assignments, is repetitive and lasts for 3,5 months.”

6. To minimize these biases, researchers could consider using multiple methods to collect data, such as surveys or focus groups, to triangulate the data and improve the validity of the findings. Additionally, researchers could employ measures to minimize bias, such as using open-ended questions to avoid leading participants to certain responses, utilizing trained interviewers who are not involved in the treatment delivery, and conducting interviews closer to the completion of treatment to reduce recall bias

Authors’ response: Thanks for this comment, we used open questions and a lot of follow up questions to further explore themes that was brought up in the interviews (page 10). Our interviewers were not involved in the research team, and they were trained and experienced psychologists (page 10). Using surveys and focus groups methodology, in addition to the interviews, would provide triangulation of data, however, we do not agree that additional methods would improve the validity of these specific findings. The reason is that to explore in-depth perceptions of participant experiences, neither surveys nor focus groups are the appropriate methods and would thus not improve the validity of findings. Validity in our study is strengthened by the highly corresponding responses in individual interviews. We have revised the text in the discussion regarding our thoughts on weaknesses in the study, see Limitations page 31.

7. Overall, this study was conducted in a thorough and thoughtful manner, generating insightful results and stimulating discussion. The results provide valuable insights into the topic factors of Importance for Change, Progress in treatment and Challenges with ADHD-I and remaining needs. The discussion section provides a nuanced and thoughtful analysis of the implications of the findings, highlighting potential areas for future research and identifying practical applications. Overall, this study is a valuable contribution to the academic literature and provides a strong foundation for further inquiry.

Authors’ response: Thank you! 

Reviewer #5: A relevant and interesting topic is discussed. The study is aimed at evaluating group therapy participants experience of a new protocol used to treat attention deficit/hyperactivity disorder inattentive presentation (ADHD-I). The researchers utilized a new cognitive-behavioral therapy for ADHD-I (CADDI). Participants provided qualitative feedback about their experience. Within this feedback themes of participants experiences were identified including factors of importance for change, gains in treatment, and challenges with ADHD and remaining needs.

This paper provides additional research supporting the development of the CADDI. Although sample size was small, the addition of the concept of information of power provides support for the study. The authors also acknowledge that a completely exhaustive data collection is rare due to limited resources. However, 

1. In regards to the sample it may be beneficial to add other characteristics of the participants. For example, education level or IQ and age participant was diagnosed with ADHD are two possible confounds. These factors may affect a participants cognitive ability to participate in the CADDI protocol.

Authors’ response: Thanks for this important suggestion, we have added more information on demographic factors (page 7). “Participants had a mean age of 32.6 years (SD=8.2, range= 22-50) and average age when first diagnosed with ADHD-I was 29.9 years (SD=9.1, range= 18-48). Nine (64%) participants were treated with stimulants with or without other pharmacotherapy, three (21 %) were medication free and two (14 %) were treated with non-stimulant medication. Regarding level of education, nine (64 %) participants had completed high school, four (29 %) had a bachelor’s or master’s degree and one (7 %) participant had not completed high school.”

2. As for materials and methods, Table 1 provides a good visualization of the protocol. The methods section could benefit from a more detailed explanation of the timeline for interviews. Was there a reasoning behind waiting three to five weeks past treatment completion? Do you think the responses would have differed if the interviews occurred within a week of completion or longer than the five weeks?

Authors’ response: Thanks for this comment. We aimed at conducting interviews as soon as possible after completing CADDI treatment. Unfortunately, scheduling difficulties in both interviewee and interviewers hindered this ambition. Some groups ended close to the summer season with vacation and traveling among the interviewees affecting scheduling of interviews. Thus, interviews were performed within a larger timeframe than we had intended. However, we believe there could be pros and cons of conducting interviews within one week or longer than five weeks. If within one week there could be a risk of recency effects; that is, that participants remember the last parts of the treatment better and thus that these parts are attributed more significance. On the other hand, one week following treatment, participants’ memory is less influenced by other factors than their experiences of treatment. If longer than five weeks, there could be a risk of memory failure and a memory influenced by other factors than treatment. On the other hand, five weeks following treatment, participants may have had time to consolidate the whole treatment experience and thus to form a more balanced experience, without recency effect bias. We do believe that the perception of treatment and recall of the 3.5 months long treatment is acceptable 3-5 weeks past completion. We have elaborated on this under Limitations in the Discussion section see page 31 “Interviews were conducted within a larger timeframe than intended due to scheduling issues in the summertime, and this might have affected recall. However, memory of CADDI is hopefully kept since the protocol includes home assignments, is repetitive and lasts for 3,5 months.” 

3. Line 105: spelling error “vent” should be “went”

Authors’ response: Thanks for this edit.

4. The inclusion criteria are addressed in lines 106 through 108. “Inclusion criteria included (i) stable use or no use of stimulations…” What is stable use of stimulants defined as? Do they have to be one the stimulants for a certain period of time?

Authors’ response: Thanks for this comment, this criterion regards the RCT and we had a 2 months period and no changes during the study defined as “stable”. We have added clearer information on the inclusion criteria in this study at page 6-7 “Eligible participants in the RCT were adults with ADHD-I aged 18 years or older. Inclusion criteria were: (i) No change in medication the last two months” 

5. The research can be used to provide first hand knowledge of the participants experiences to future clients interested in the protocol. Clients often ask “why do group therapy?” or “how is this going to help me?” This research answers those common questions.

Authors’ response: Thank you, we share this opinion.

6. Treatment for ADHD seems to be moving toward a multimodal approach. Future research with this protocol could benefit from looking at the effect of prescribed medications along with therapy versus therapy only.

Authors’ response: Thanks for this comment, it´s an interesting question and we might look into that in a future study, as this question is a little besides the scoop of this study. 

Reviewer #6: 

1. The manuscript is based on impressive empirical evidence and makes an original contribution, but findings cane be questioned because CADDI protocol itself is still in its initial stage of development and not yet proven standard effective method of therapy.

Reporting bias of the participants should also be commented and method to reduce it.

Authors’ response: Thanks for this comment. We used independent interviewers, and we invited all participants to report on positive and negative experiences of CADDI. We also used independent transcribers. We have added more information regarding this in the manuscript (page 10). “The interviewers were independent of the research group and this was explicitly stated by the interviewers before the interview started. The interviewers had no previous relationship with participants, to allow participants to express themselves freely, without consideration or perceived need to please the interviewer. The interviews were transcribed verbatim by assistants with little previous knowledge about CADDI and not involved in the analysis of data, to ensure transcriptions would not be influenced by preconceptions of the study objectives.” Further, we were two authors with different perspectives doing the analysis to reduce bias in the analyzing phase which is now more explicitly stated (page 13). “Thus, we choose researchers who brought different perspectives to the analysis; EES, MSc, is a licensed clinical psychologist and psychotherapist, specialized in CBT for ADHD. She has developed the CADDI protocol and is currently involved in conducting a multicenter trial of the protocol. In addition, EES has been involved as a group leader of CADDI and supervisor of group leaders in the groups participating in the RCT. RS, PhD, is a licensed clinical psychologist with no former knowledge of the CADDI protocol and no experience of CBT for ADHD. Thus, the different perspectives that EES and RS brought to the analysis, one involved and the other naïve, were considered beneficial to the analysis of data.” 

2. Results section should be shortened to improve the readability of the manuscript (may be eliminate the individual sentence response from participants as it doesn't require from my perspective).

Authors response: Thanks for this comment, we have revised the Result section and shortened it by eliminating individual sentence responses and kept more space for elaborate responses. 

3. Otherwise, its a very good and novel type of article and will be very interesting to see if this CADDI protocol becomes standard of CBT for ADD

Authors response: Thank you! 

Reviewer #7: PONE-D-23-32070

Thank you for the opportunity to review this manuscript “It was very nice to be in a room where everyone had ADD - that’s kind of VIP” Clients’. Perspectives on Group CBT for ADHD Inattentive Presentation.” The study aimed at capture 1) What aspects of treatment the participants found to be helpful, and 2) If there were areas that ought to be developed to make the protocol more useful to clients with ADHD-I. The authors designed a study with data collection using an interview protocol with semi-structured questions and invited individuals who had received treatment based on the CADDI protocol. The interview data was analysed with thematic analysis described by Braun and Clarke. It is a highly relevant topic and when designing therapeutic interventions, it´s always of value to have participants experiences.

1. Line 93. 

You saying that you aiming at explore participants’ perceptions of CADDI. In the

next sentence the concept: lived experience, is used. This is not in line with exploring perceptions with a semi-structured interview-guide. My suggestion: remove the sentence.

Also, the aim in the abstract, line 33,: …explored participants' experiences of … is not in line with the aim in the text, line 94, explore participants’ perceptions of…. Please use the same wording in the abstract and in the introduction and in the Discussion, line 455: explore participants’ perceptions.

Response: Thank you for pointing out this, this has been corrected in the text. According to your suggestion we have how removed the word ”lived experiences” and ”experiences” throughout the manuscript and now only use ”participant’s perceptions”.

2. Line 130. 

Participants were treated in three groups at separate locations, group-size ranged from 5 to 6 participants. What was the reasons for the relatively small groups, the group session was designed for 6-10 participants?

Authors’ response: In this study our sample was recruited from a RCT that was conducted partly during the pandemic. During the pandemic we had limited group sizes allowing groups of six as a maximum. In some of the locations we had difficulties recruiting participants as needed for larger groups. This information has been added to the text, page 11 “Participants were treated in three separate groups, with group size ranging from 5 to 6 participants, due to restrictions on group size during the Covid-19 pandemic or limited base of recruitment at some locations”

3. Line 150. 

Interviews were performed by three licensed clinical psychologists trained in CBT. 

Why is their competence made clear? There is no explanation here or in the methodological considerations.

Authors’ response: Thank you, we consider knowledge in the field of CBT important for the interviewer to perceive information given in the interviews regarding treatment in CADDI, since CADDI is a CBT protocol. We have added a brief explanation of this at page 12: ”Interviews were performed by three licensed clinical CBT psychologists, familiar with adult ADHD, to understand treatment and thus participant responses.”

4. Line 166

The analysis was primarily inductive….

This is something that I can´t see in your result. Using questions such as: “How has the treatment affected you?”, “What components of the treatment were most helpful to you?”, “Is there anything that could have made this treatment more helpful to you?” and then end up with three themes: Factors of importance for change, Gains in treatment and

Challenges with ADHD-I and remaining needs.

Were these themes identified inductively? That´s difficult to see. When you have structured questions (to some degree) with specific questions in a relatively limited question area, it´s evidently very difficult to use inductive analysis. That may be that you used induction within the search for subthemes? Please rephrase this in the Data analysis.

Authors’ response: This is a very important question regarding our method, and we agree that it may seem as we were simply categorizing answers to the questions, but that is not how we did the analysis. This is important and therefore we have expanded on the description of how the analysis was conducted in the manuscript. We followed the six phases of reflexive thematic analysis (RTA) and searched the dataset for codes, themes and from there organized our observations in subthemes and main themes. The themes were identified from the codes that were generated from the text, the text was not analyzed through a theoretical framework i.e., deductively. We were indeed interested in all findings and not restricted to those corresponding to the research question as is reflected for instance in the main theme “Challenges with ADHD-I and remaining needs”. We have elaborated the description of the analysis process in the text at page 11-13. 

5. Line 168. 

You are using “information power;” as a pragmatic saturation, and is referring to Braun Clarke. Please use the original author of this: Malterud K, Siersma VD, Guassora AD. Sample Size in Qualitative Interview Studies: Guided by Information Power. Qual Health Res. 2016 Nov;26(13):1753-1760. doi: 10.1177/1049732315617444. Epub 2016 Jul 10. PMID: 26613970.

Authors’ response: Thank you! We have revised the text and use the concept of pragmatic saturation through the manuscript. Pragmatic saturation is close to information power but more often used in thematic analysis. See page 12: “Sample size was determined by applying the concept of “pragmatic saturation”, assuming that data collection is sufficient when the data give rich and multifaceted information regarding the research questions and new data does not contribute to new themes being generated (35).”

6. Themes and subthemes

The analysis is surely well conducted and the subthemes seems to belong to their respective theme. But, naming the themes can be challenging and I don´t think that you have taking the time and discussions to do so. Now your themes more resemble purely grouping the statements. In describing the analysis, line 168, you are so nicely expressing: …considering verbal statements to be reflections of personal experience and meaning. Please change and/or develop naming of the themes, and some of the subthemes, so the result for us readers will be more descriptive and more in line with the participants experiences and meanings.

Authors’ response: Thank you, as described above we followed the steps of analysis according to Braun Clarke and we have elaborated on this process in the text (pages 11-12). We are trying to be both descriptive, as in the texts and quotes supporting each theme/subtheme, and accurately defining headings to make the results accessible to readers. Naming themes and subthemes is indeed challenging, and we might have gone to a more technical and less vivid style in doing so, in the name of themes clarity and accuracy of the content of the themes. In the process of analyzing data, naming the themes is the ultimate step and is preceded by many other phases where the results have been processed and this work led to the naming of our themes. As described in Braun Clarke (2006) we choose these names to best describe the content of themes they represent. 

7. I don’t recognize any discussions of limitations or methodological considerations.

Please complete the Discussion with your thoughts of strengths and weaknesses.

Authors response: Thank you, that is a good suggestion, we have added sections regarding this “Limitations” and “Strengths of the study” and “Implications for practice and future research”. See pages 27-33.

Reviewer #8: Thank you for the opportunity to review this manuscript, which I enjoyed reading. It is well-written and describe thoroughly both methods used and the results from the qualitative interviews. It is important to listen to adults with ADHD on how they experience treatment interventions and not just exclusively use quantitative self-reports. I found the manuscript to be methodically of high quality. However, I am not an expert on the use of qualitative methods and may have overlooked methodological aspects of the manuscript that would be beneficial to improve.

1. I am one of the co-authors of the study of Nordby et al. (2021) and will as such make a note that this study was not a CBT but rather a neurocognitive training intervention in a group-setting. From reading the description of the CBT protocol used in the submitted study, I can clearly see that it is overlaps with the Goal Management Training (GMT) used in Nordby et al. But still GMT is not reckoned as a therapeutic intervention but rather a training intervention. This is an important distinction to make and relevant for the different findings discussed in the discussion section. It is interesting that across different types of interventions, adults with ADHD experience a group setting as beneficial and helpful by meeting other adults with similar difficulties. However, the training on executive functions in GMT may tap into the ADHD symptoms more compared with CBT by higher requirements on attentional control. This could be one of the reasons for why it was a link between heterogeneity in ADHD symptoms and feelings of exclusion in the Nordby study – such as that the training it-self will require a greater ability to be attentive and being less restless/fidgeting. As such, more severe ADHD symptoms can make the training on executive functions more challenging and give a feeling of not mastering the training as others in the same intervention group. I believe adding more on the distinction of the interventions and discuss differences in findings will enrich the discussion section and make it more relevant for the readers working clinically with or doing research on ADHD.

Authors’ response: Thank you, it´s interesting to hear more about the Nordby study. We have revised the text to refer correctly to your study. See pages 27 “The experience of being one among many others with ADHD-I was described to ease the burden of shame, self-criticism, and loneliness, an effect that has been observed in previous studies of group interventions for ADHD (37, 38)”, and page 28 “Ambiguity regarding the impact of the group was reported in a study by Nordby and colleagues (38), where participants in “Goal management training” perceived the group as both supporting and distracting the work in the group. Heterogeneity regarding ADHD symptoms in the group caused feelings of exclusion in some participants thus pointing to challenges in group settings with various symptoms (38).” Further, we would like to discuss the distinctions of the interventions, but unfortunately, we found this difficult as we didn’t find enough information regarding the components of Goal Management Training to make a just comparison with it in our Discussion. 

2. It is further very interesting that focusing on ADHD-I may have beneficial effects on the group setting of CBT. However, I think the authors could discuss the beneficial effects of focusing on ADHD subgroups against the general trend in intervention research to focus on transdiagnostic factors. It is not realistic in a clinical setting to have too many different interventions targeting specific mental health subgroups. This is one of the reasons for why there is an increased focus on developing and testing the effects of transdiagnostic interventions.

Authors’ response: Thanks for this comment. We have developed the discussion and point on the possible importance of homogeneity for the therapeutic effects of the groups as this is a finding in our study. The topic of focusing subgroups or transdiagnostic factors when developing new treatment is very important and interesting. The pros and cons regarding specific versus transdiagnostic interventions deserves more attention and research. However, this issue is beyond the scoop of our study.

Reviewer #9: 

1. The article is wordy and can be edited to be more concise. Authors can consider formatting article in a clear sections, such as, limitation, and gaps in knowledge. Authors can consider elaborating if 14 candidates had 100% attendance; were any participants that was removed from study? 

Authors’ response: Thank you for these suggestions. We have revised the text to be more concise (shortening the Result section by eliminating redundant quotes) and described relevant information more clearly, using subheadings to structure the paper. We have developed the Discussion with subheadings, “Strengths of the study”, “Limitations” and “Implications for practice and future research” where we further elaborate our thought on these questions, see pages 29-33.

Regarding participants, we invited 15 persons from three treatment groups to participate, of those 14 accepted to be interviewed. No participants were removed from the study. We have elaborated the text on how participants were recruited at page 6-7: “Eligible participants in the RCT were adults with ADHD-I aged 18 years or older. Inclusion criteria were: (i) No change in medication two months prior to inclusion and (ii) completion of a psychoeducational course on ADHD focusing on symptoms, self-care and treatments. Exclusion criteria were: (i) severe mental illness (e.g., severe depression), (ii) substance abuse, or (iii) intellectual disability. Additional inclusion criteria for the current interview study were: (i) randomized to CADDI and (ii) completion (i.e. not dropping out) of the CADDI intervention.”

2. The interviews were supervised by EES but it would be helpful to spell out that the structural interview was standardized.

Authors’ response: The interviews were not supervised by EES. The interviews were conducted using a semi structured guide, we have expanded the description of the interview guide and how it was developed. See page 10; ”A semi-structured interview guide was developed by our research team through discussions regarding the study’s objective and through two pilot interviews with individuals who had completed CADDI but were not part of the sample in this study. The interview guide included questions concerning participants’ perceptions of CADDI. Initially some general questions were posed to start the conversation and to facilitate recall of the CADDI protocol, for example, “What was the overall aim of this treatment?”, “What was the content of this treatment?” Thereafter more personal question regarding perceptions of treatment and its effect were asked; “How has the treatment affected you?”, “What components of the treatment were most helpful to you?”, Ultimately, questions regarding evaluation of treatment were asked; “Is there anything that could have made this treatment more helpful to you?”, “Is there anything that was missing in treatment or that you would have liked to see more of? and “Did you experience any negative effects from treatment?”. 

3. Line 106-107 can authors elaborate "stable use." 

Authors’ response: We have rewritten the text as follows at page 6-7: “ Inclusion criteria were: (i) No change in medication the last two months and (ii) completion of a psychoeducational course on ADHD regarding symptoms, self-care and treatments.” 

4. Also, it was unclear if participants were treated with non stimulant ADHD medication

Authors’ response: We have added information on medication status as proposed, in the section regarding participants at page 7; “Nine (64 %) participants were treated with stimulants with or without other pharmacotherapy, three (21 %) were medication free and two (14 %) were treated with non-stimulant medication.” 

5. Line 108 authors can consider describing "severe mental illness."

Author’s response: We have rewritten this to be more precise (page 7): “Exclusion criteria were, (i) severe mental illness (e.g., severe depression)”

---

## [Decision Letter · Decision Letter 1]

5 Feb 2024

“It was very nice to be in a room where everyone had ADD - that’s kind of VIP”: Exploring Clients’ Perceptions of Group CBT for ADHD Inattentive Presentation

PONE-D-23-32070R1

Dear Dr. Strålin,

We’re pleased to inform you that your manuscript has been judged scientifically suitable for publication and will be formally accepted for publication once it meets all outstanding technical requirements.

Kind regards,

Lakshit Jain, MD

Academic Editor

PLOS ONE

Additional Editor Comments (optional):

Reviewers' comments:

Reviewer's Responses to Questions

**Comments to the Author**

1. If the authors have adequately addressed your comments raised in a previous round of review and you feel that this manuscript is now acceptable for publication, you may indicate that here to bypass the “Comments to the Author” section, enter your conflict of interest statement in the “Confidential to Editor” section, and submit your "Accept" recommendation.

Reviewer #1: All comments have been addressed

Reviewer #3: All comments have been addressed

Reviewer #6: All comments have been addressed

Reviewer #8: All comments have been addressed

2. Is the manuscript technically sound, and do the data support the conclusions?

Reviewer #1: Yes

Reviewer #3: (No Response)

Reviewer #6: Partly

Reviewer #8: Yes

3. Has the statistical analysis been performed appropriately and rigorously? 

Reviewer #1: Yes

Reviewer #3: (No Response)

Reviewer #6: I Don't Know

Reviewer #8: N/A

4. Have the authors made all data underlying the findings in their manuscript fully available?

Reviewer #1: Yes

Reviewer #3: (No Response)

Reviewer #6: Yes

Reviewer #8: No

5. Is the manuscript presented in an intelligible fashion and written in standard English?

Reviewer #1: Yes

Reviewer #3: (No Response)

Reviewer #6: Yes

Reviewer #8: Yes

6. Review Comments to the Author

Reviewer #1: The revisions to your paper have been diligently implemented, resulting in a fully corrected and now acceptable manuscript. We extend our gratitude for your dedicated efforts and collaborative approach in elevating the overall quality of your article.

We wish you continued success in your forthcoming research and writing endeavors.

Reviewer #3: All the concerns that were raised at the time of review were addressed by author. The manuscript is acceptable for publication.

Reviewer #6: (No Response)

Reviewer #8: Thank you for the responses to my comments. I think the manuscript is methodologically sound and of interest for other researchers and clinicians. .

7. PLOS authors have the option to publish the peer review history of their article (what does this mean?). If published, this will include your full peer review and any attached files.

Reviewer #1: **Yes: **Roghieh Nooripour

Reviewer #3: **Yes: **Jasleen Kaur MD

Reviewer #6: No

Reviewer #8: **Yes: **Lin Sørensen

---

## [Editor Report · Acceptance letter]

26 Mar 2024

PONE-D-23-32070R1 

PLOS ONE

Dear Dr. Strålin, 

I'm pleased to inform you that your manuscript has been deemed suitable for publication in PLOS ONE. Congratulations! Your manuscript is now being handed over to our production team.

Kind regards, 

on behalf of

Dr. Lakshit Jain 

Academic Editor

PLOS ONE